# CONTROLLING DIRECTIONS ORTHOGONAL TO A CLASSIFIER

**Yilun Xu, Hao He, Tianxiao Shen, Tommi Jaakkola**
Computer Science and Artificial Intelligence Lab,
Massachusetts Institute of Technology
{ylxu, haohe, tianxiao}@mit.edu; tommi@csail.mit.edu

## ABSTRACT

We propose to identify directions invariant to a given classifier so that these directions can be controlled in tasks such as style transfer. While orthogonal decomposition is directly identifiable when the given classifier is linear, we formally define a notion of orthogonality in the non-linear case. We also provide a surprisingly simple method for constructing the orthogonal classifier (a classifier utilizing directions other than those of the given classifier). Empirically, we present three use cases where controlling orthogonal variation is important: style transfer, domain adaptation, and fairness. The orthogonal classifier enables desired style transfer when domains vary in multiple aspects, improves domain adaptation with label shifts and mitigates the unfairness as a predictor. The code is available at https://github.com/Newbeeer/orthogonal_classifier.

## 1 INTRODUCTION

Many machine learning applications require explicit control of directions that are orthogonal to a predefined one. For example, to ensure fairness, we can learn a classifier that is orthogonal to sensitive attributes such as gender or race (Zemel et al., 2013; Madras et al., 2018). Similar, if we transfer images from one style to another, content other than style should remain untouched. Therefore images before and after transfer should align in directions orthogonal to style. Common to these problems is the task of finding an orthogonal classifier. Given any *principal classifier* operating on the basis of *principal variables*, our goal is to find a classifier, termed *orthogonal classifier*, that predicts the label on the basis of *orthogonal variables*, defined formally later.

The notion of orthogonality is clear in the linear case. Consider a joint distribution $P_{XY}$ over $X \in \mathbb{R}^d$ and binary label $Y$. Suppose the label distribution is Bernoulli, *i.e.*, $P_Y = \mathcal{B}(Y; 0.5)$ and class-conditional distributions are Gaussian, $P_{X|Y=y} = \mathcal{N}(X; \mu_y, \sigma_y^2 I)$, where the means and variances depend on the label. If the principal classifier is linear, $w_1 = \Pr(Y = 1|\theta_1^\top x)$, any classifier $w_2$, in the set $\mathcal{W}_2 = \{\Pr(Y = 1|\theta_2^\top x) \mid \theta_1^\top \theta_2 = 0\}$, is considered orthogonal to $w_1$. Thus the two classifiers $w_1, w_2$, with orthogonal decision boundaries (Fig. 1) focus on distinct but complementary attributes for predicting the same label.

Finding the orthogonal classifier is no longer straightforward in the non-linear case. To rigorously define what we mean by the orthogonal classifier, we first introduce the notion of mutually orthogonal random variables that correspond to (conditinally) independent latent variables mapped to observations through a diffeomorphism (or bijection if discrete). Each r.v. is predictive of the label but represents complementary information. Indeed, we show that the orthogonal random variable maximizes the conditional mutual information with the label given the principal counterpart, subject to an independence constraint that ensures complementarity.

Our search for the orthogonal classifier can be framed as follows: given a principal classifier $w_1$ using some unknown principal r.v. for prediction, how do we find its orthogonal classifier $w_2$ relying solely on orthogonal random variables? The solution to this problem, which we call *classifier orthogonalization*, turns out to be surprisingly simple. In addition to the principal classifier, we assume access to a full classifier $w_x$ that predicts the same label based on all the available information, implicitly relying on both principal and orthogonal latent variables. The full classifier can be trained normally, absent of constraints[1]. We can then effectively "subtract" the contribution of $w_1$ from the full classifier to obtain the orthogonal classifier $w_2$ which we denote as $w_2 = w_x \smallsetminus w_1$. The advantage of this construction is that

---

[1] The full classifier may fail to depend on all the features, e.g., due to simplicity bias (Shah et al., 2020).

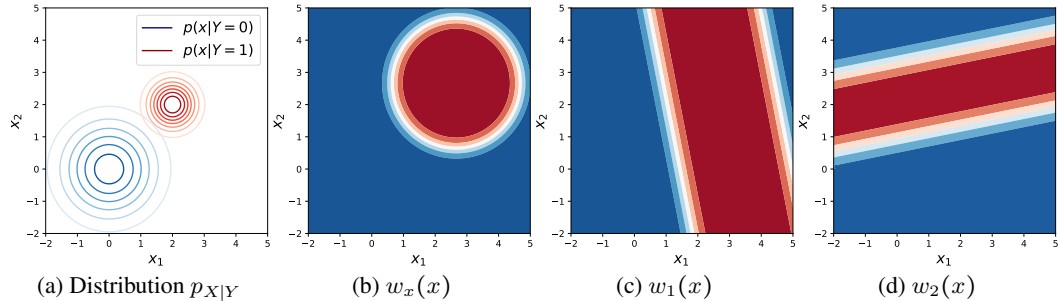

(a) Distribution $p_{X|Y}$     (b) $w_x(x)$     (c) $w_1(x)$     (d) $w_2(x)$

Figure 1: An illustrative example of orthogonal classifier in a linear case. (a) is the data distributions in two classes; (b,c,d) are the probabilities of the data from class 1 predicted by the full/principal/orthogonal classifiers. Red and blue colors mean a probability close to 1 or 0. The white color indicates regions with a probability close to $0.5$, which are classifiers' decision boundaries. Clearly, $w_1$ and $w_2$ have orthogonal decision boundaries.

we do not need to explicitly identify the underlying orthogonal variables. It suffices to operate only on the level of classifier predictions.

We provide several use cases for the orthogonal classifier, either as a predictor or as a discriminator. As a predictor, the orthogonal classifier predictions are invariant to the principal sensitive r.v., thus ensuring fairness. As a discriminator, the orthogonal classifier enforces a partial alignment of distributions, allowing changes in the principal direction. We demonstrate the value of such discriminators in 1) *controlled style transfer* where the source and target domains differ in multiple aspects, but we only wish to align domain A's style to domain B, leaving other aspects intact; 2) *domain adaptation with label shift* where we align feature distributions between the source and target domains, allowing shifts in label proportions. Our results show that the simple method is on par with the state-of-the-art methods in each task.

## 2    NOTATIONS AND DEFINITION

**Symbols.** We use the uppercase to denote random variable (*e.g.*, data $X$, label $Y$), the lowercase to denote the corresponding samples and the calligraphic letter to denote the sample spaces of r.v., *e.g.*, data sample space $\mathcal{X}$. We focus on the setting where label space $\mathcal{Y}$ is discrete, *i.e.*, $\mathcal{Y} = \{1, \cdots, C\}$, and denote the $C - 1$ dimensional probability simplex as $\Delta^C$. A classifier $w : \mathcal{X} \to \Delta^C$ is a mapping from sample space to the simplex. Its $y$-th dimension $w(x)_y$ denotes the predicted probability of label $y$ for sample $x$.

**Distributions.** For random variables $A, B$, we use the notation $p_A$, $p_{A|B}$, $p_{AB}$ to denote the marginal/conditional/joint distribution, *i.e.*, $p_A(a) = p(A = a)$, $p_{A|B}(a|b) = p(A = a|B = b)$, $p_{AB}(a, b) = p(A = a, B = b)$. Sometimes, for simplicity, we may ignore the subscript if there is no ambiguity, *e.g.*, $p(a|b)$ is an abbreviation for $p_{A|B}(a|b)$.

We begin by defining the notion of an orthogonal random variable. We consider continuous $X, Z_1, Z_2$ and assume their supports are manifolds diffeomorphic to the Euclidean space. The probability density functions (PDF) are in $\mathcal{C}^1$. Given a joint distribution $p_{XY}$, we define the orthogonal random variable as follows:

**Definition 1** (Orthogonal random variables). *We say $Z_1$ and $Z_2$ are orthogonal random variables w.r.t $p_{XY}$ if they satisfy the following properties:*

*(i) There exists a diffeomorphism $f : \mathcal{Z}_1 \times \mathcal{Z}_2 \to \mathcal{X}$ such that $f(Z_1, Z_2) = X$.*

*(ii) $Z_1$ and $Z_2$ are statistically independent given $Y$, i.e., $Z_1 \perp\!\!\!\perp Z_2|Y$.*

The orthogonality relation is symmetric by definition. Note that the orthogonal pair perfectly reconstructs the observations via the diffeomorphism $f$; as random variables they are also sampled independently from class conditional distributions $p(Z_1|Y)$ and $p(Z_2|Y)$. For example, we can regard foreground objects and background scenes in natural images as being mutually orthogonal random variables.

**Remark.** The definition of orthogonality can be similarly developed for discrete variables and discrete-continuous mixtures. For discrete variables, for example, we can replace the requirement of diffeomorphism with bijection.

Since the diffeomorphism $f$ is invertible, we can use $z_1 : \mathcal{X} \to \mathcal{Z}_1$ and $z_2 : \mathcal{X} \to \mathcal{Z}_2$ to denote the two parts of the inverse mapping so that $Z_1 = z_1(X)$ and $Z_2 = z_2(X)$. Note that, for a given joint distribution $p_{XY}$, the decomposition into orthogonal random variables is not unique. There are multiple pairs of random variables that represent valid mutually orthogonal latents of the data. We can further justify our definition of orthogonality from an information theoretic perspective by showing that the choice of $z_2$ attains the maximum of the following constrained optimization problem.

**Proposition 1.** *Suppose the orthogonal r.v. of $z_1(X)$ w.r.t $p_{XY}$ exists and is denoted as $z_2(X)$. Then $z(X) = z_2(X)$ is a maximizer of $I(z(X); Y | z_1(X))$ subject to $I(z(X); z_1(X) | Y) = 0$.*

We defer the proof to Appendix B.1. Proposition 1 shows that the orthogonal random variable maximizes the additional information about the label we can obtain from $X$ while remaining conditionally independent of the principal random variable. This ensures complementary in predicting the label.

## 3    CONSTRUCTING THE ORTHOGONAL CLASSIFIER

Let $Z_1 = z_1(X)$ and $Z_2 = z_2(X)$ be mutually orthogonal random variables w.r.t $p_{XY}$. We call $Z_1$ the principal variable and $Z_2$ the orthogonal variable. In this section, we describe how we can construct the Bayes optimal classifier operating on features $Z_2$ from the Bayes optimal classifier relying on $Z_1$. We formally refer to the classifiers of interests as: (1) principal classifier $w_1(x)_y = p(Y = y | Z_1 = z_1(x))$; (2) orthogonal classifier $w_2(x)_y = p(Y = y | Z_2 = z_2(x))$; (3) full classifier $w_x(x)_y = p(Y = y | X = x)$.

### 3.1    CLASSIFIER ORTHOGONALIZATION

Our key idea relies on the bijection between the density ratio and the Bayes optimal classifier (Sugiyama et al., 2012). Specifically, the ratio of densities $p_{X|Y}(x|i)$ and $p_{X|Y}(x|j)$, assigned to an arbitrary point $x$, can be represented by the Bayes optimal classifier $w(x)_i = \Pr(Y = i|x)$ as $\frac{p_{X|Y}(x|i)}{p_{X|Y}(x|j)} = \frac{p_Y(j)w(x)_i}{p_Y(i)w(x)_j}$. Similar, the principal classifier $w_1$ gives us associated density ratios of class-conditional distributions over $Z_1$. For any $i, j \in \mathcal{Y}$, we have $\frac{p_{Z_1|Y}(z_1(x)|i)}{p_{Z_1|Y}(z_1(x)|j)} = \frac{p_Y(j)w_1(x)_i}{p_Y(i)w_1(x)_j}$. These can be combined to obtain density ratios of class-conditional distribution $p_{Z_2|Y}$ and subsequently calculate the orthogonal classifier $w_2$. We additionally rely on the fact that the diffeomorphism $f$ permits us to change variables between $x$ and $z_1, z_2$: $p_{X|Y}(x|i) = p_{Z_1|Y}(z_1|i) * p_{Z_2|Y}(z_2|i) * \text{vol} J_f(z_1, z_2)$, where $\text{vol} J_f$ is volume of the Jacobian (Berger et al., 1987) of the diffeomorphism mapping. Taken together,

$$\frac{p_{Z_2|Y}(z_2|i)}{p_{Z_2|Y}(z_2|j)} = \frac{p_{Z_1|Y}(z_1|i) * p_{Z_2|Y}(z_2|i) * \text{vol} J_f(z_1, z_2)}{p_{Z_1|Y}(z_1|j) * p_{Z_2|Y}(z_2|j) * \text{vol} J_f(z_1, z_2)} \bigg/ \frac{p_{Z_1|Y}(z_1|i)}{p_{Z_1|Y}(z_1|j)} = \frac{w_x(x)_i}{w_x(x)_j} \bigg/ \frac{w_1(x)_i}{w_1(x)_j} \quad (1)$$

Note that since the diffeomorphism $f$ is shared with all classes, the Jacobian is the same for all label-conditioned distributions on $Z_1, Z_2$. Hence the Jacobian volume terms cancel each other in the above equation. We can then finally work backwards from the density ratios of $p_{Z_2|Y}$ to the orthogonal classifier:

$$\Pr(Y = i | Z_2 = z_2(x)) = \Pr(Y = i) \frac{w_x(x)_i}{w_1(x)_i} \bigg/ \sum_j \left( \Pr(Y = j) \frac{w_x(x)_j}{w_1(x)_j} \right) \quad (2)$$

We call this procedure *classifier orthogonalization* since it adjusts the full classifier $w_x$ to be orthogonal to the principal classifier $w_1$. The validity of this procedure requires overlapping supports of the class-conditional distributions, which ensures the classifier outputs $w_x(x)_i, w_1(x)_i$ to remain non-zero for all $x \in \mathcal{X}, i \in \mathcal{Y}$.

Empirically, we usually have access to a dataset $\mathcal{D} = \{(x_t, y_t)\}_{t=1}^n$ with $n$ iid samples from the joint distribution $p_{XY}$. To obtain the orthogonal classifier, we need to first train the full classifier $\hat{w}_x$ based on the dataset $\mathcal{D}$. We can then follow the classifier orthogonalization to get an empirical orthogonal classifier, denoted as $w_2 = \hat{w}_x \setminus w_1$. We use symbol $\setminus$ to emphasize that the orthogonal classifier uses complementary information relative to $z_1$. Algorithm 1 summarizes the construction of the orthogonal classifier.

**Generalization bound.** Since $w_x$ is trained on a finite dataset, we consider the generalization bound of the constructed orthogonal classifier. We denote the population risk as $R(w) = -\mathbb{E}_{p_{XY}(x,y)} \left[ \log w(x)_y \right]$ and the empirical risk as $\hat{R}(w) = -\frac{1}{|\mathcal{D}|} \sum_{(x_i, y_i) \in \mathcal{D}} \log w(x_i)_{y_i}$. For a function family $\mathcal{W}$ whose elements map $\mathcal{X}$ to the simplex $\Delta^C$, we define $\hat{w}_x = \inf_{w_x \in \mathcal{W}} \hat{R}(w_x), w_x^* = \inf_{w_x \in \mathcal{W}} R(w_x)$. We further denote the Rademacher complexity of function family $\mathcal{W}$ with sample size $|\mathcal{D}|$ as $\mathfrak{R}_{|\mathcal{D}|}(\mathcal{W})$.

---

**Algorithm 1** Classifier Orthogonalization

---

**Input:** principal classifier $w_1$, dataset $\mathcal{D}$.
Train an empirical full classifier $\hat{w}_x$ on $\mathcal{D}$ by empirical risk minimization.
Construct an orthogonal classifier $\hat{w}_x \smallsetminus w_1$ via classifier orthogonalization (Eq. (2)).
**return** the empirical orthogonal classifier $\hat{w}_2 = \hat{w}_x \smallsetminus w_1$

---

**Theorem 1.** *Assume $p_y$ is uniform distribution, $\forall w_x \in \mathcal{W}$ takes values in $(m, 1-m)^C$ with $m \in (0, \frac{1}{2})$, and $1/p_{X|Y}(x|y) \in (0,\gamma) \subset (0,+\infty)$ holds for $\forall x \in \mathcal{X}, y \in \mathcal{Y}$. Then for any $\delta \in (0,1)$ with probability at least $1-\delta$, we have:*

$$|R(\hat{w}_x \smallsetminus w_1) - R(w_x^* \smallsetminus w_1)| \leq (1+\gamma)\left(2\mathfrak{R}_{|\mathcal{D}|}(\mathcal{W}) + 2\log\frac{1}{m}\sqrt{\frac{2\log\frac{1}{\delta}}{|\mathcal{D}|}}\right)$$

Theorem 1 shows that the population risk of the empirical orthogonal classifier in Algorithm 1 would be close to the optimal risk if the maximum value of the reciprocal of class-conditioned distributions $1/p_{X|Y}(x|y)$ and the Rademacher term are small. Typically, the Rademacher complexity term satisfies $\mathfrak{R}_{|\mathcal{D}|}(\mathcal{W}) = \mathcal{O}(|\mathcal{D}|^{-\frac{1}{2}})$ (Bartlett & Mendelson, 2001; Gao & Zhou, 2016). We note that the empirical full classifier may fail in specific ways that are harmful for our purposes. For example, the classifier may not rely on all the key features due to simplicity bias as demonstrated by (Shah et al., 2020). There are several ways that this effect can be mitigated, including Ross et al. (2020); Teney et al. (2021).

### 3.2 ALTERNATIVE METHOD: IMPORTANCE SAMPLING

An alternative way to get the orthogonal classifier is via importance sampling. For each sample point $(x,y)$, we construct its importance $\phi(x,y) := \frac{p_{Z_1}(z_1(x))}{p_{Z_1|Y}(z_1(x)|y)}$. Via Bayes' rule, the importance $\phi(x,y)$ can be calculated from the principal classifier by $\phi(x,y) = \frac{p_Y(y)}{w_1(x)_y}$. We define the importance sampling (IS) objective as $\mathcal{L}_{\mathtt{IS}}(w) := \mathbb{E}_{x,y \sim p_{XY}}[\phi(x,y)\log w(x)_y]$. It can be shown that the orthogonal classifier $w_2$ maximizes the IS objective, *i.e.*, $w_2 = \arg\max \mathcal{L}_{\mathtt{IS}}$. However, the importance sampling method has an Achilles' heel: variance. A few bad samples with large weights can drastically deviate the estimation, which is problematic for mini-batch learning in practice. We provide an analysis of the variance of the IS objective. It shows the variance increases with the divergences between $Z_1$'s marginal and its label-conditioned marginals, $\{D_f(p_{Z_1}\|p_{Z_1|Y=y})\}_{y=1}^C$ even when the learned classifier $w$ is approaching the optimal classifier, *i.e.*, $w \approx w_2$. Let $\widehat{\mathcal{L}_{\mathtt{IS}}^n}(w) := \frac{1}{n}\sum_{t=1}^n \phi(x_t,y_t)\log w(x_t)_{y_t}$ be the empirical IS objective estimated with $n$ iid samples. Clearly, $\mathrm{Var}(\widehat{\mathcal{L}_{\mathtt{IS}}^n}(w)) = \frac{1}{n}\mathrm{Var}(\widehat{\mathcal{L}_{\mathtt{IS}}^1}(w))$. While $\mathrm{Var}(\widehat{\mathcal{L}_{\mathtt{IS}}^1}(w))$ at $w = w_2$ is the following,

$$\mathrm{Var}(\widehat{\mathcal{L}_{\mathtt{IS}}^1}(w_2)) = \mathbb{E}_Y\left[\left(D_f(p_{Z_1}\|p_{Z_1|Y=y})+1\right)\mathbb{E}_{Z_2|Y=y}\log^2 p_{Y|Z_2}(y|z_2)\right] - \mathcal{L}(w_2)^2$$

where $D_f$ is the Pearson $\chi^2$-divergence. The expression indicates that the variance grows linearly with the expected divergence. In contrast, the divergences have little effect when training the empirical full classifier in classifier orthogonalization. We provide further experimental results in section 4.1.2 to demonstrate that classifier orthogonalization has better stability than the IS objective with increasing divergences and learning rates.

## 4 ORTHOGONAL CLASSIFIER FOR CONTROLLED STYLE TRANSFER

In style transfer, we have two domains $A, B$ with distributions $P_{XY}, Q_{XY}$. We use binary label $Y$ to indicate the domain of data $X$ (0 for domain $A$, 1 for domain $B$). Assume $Z_1$ and $Z_2$ are mutually orthogonal variables w.r.t both $P_{XY}$ and $Q_{XY}$. The goal is to conduct controlled style transfer between the two domains. By controlled style transfer, we mean transferring domain $A$'s data to domain $B$ only via making changes on partial latent ($Z_1$) while not touching the other latent ($Z_2$). Mathematically, we aim to transfer the latent distribution from $P_{Z_1}P_{Z_2}$ to $Q_{Z_1}P_{Z_2}$. This task cannot be achieved by traditional style transfer algorithms such as CycleGAN (Zhu et al., 2017), since they directly transfer data distributions from $P_X$ to $Q_X$, or equivalently from the perspective of latent distributions, from $P_{Z_1}P_{Z_2}$ to $Q_{Z_1}Q_{Z_2}$. Below we show that the orthogonal classifier $w_x \smallsetminus w_1$ enables such controlled style transfer.

Our strategy is to plug the orthogonal classifier into the objective of CycleGAN. The original Cycle-GAN objective defines a min-max game between two generators/discriminator pairs $G_{AB}/D_{AB}$ and

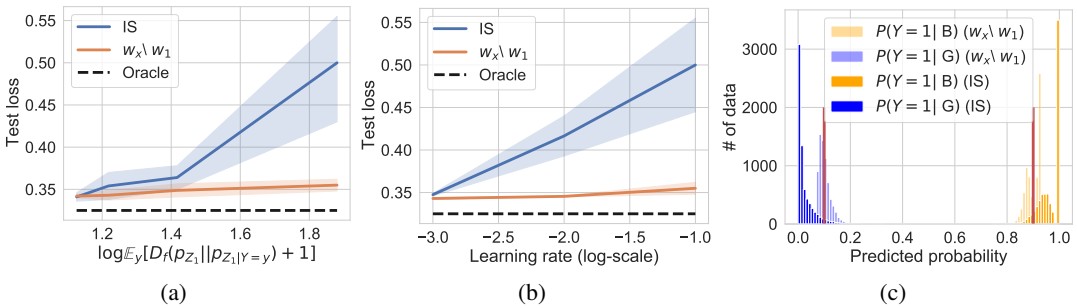

Figure 2: **(a,b):** Test loss versus log expected divergence and learning rate. **(c):** The histogram of predicted probability of different methods. The two red lines are the ground-truth probabilities for $P(Y = 1|\text{Brown background})$ and $P(Y = 1|\text{Green background})$.

$G_{BA}/D_{BA}$. $G_{AB}$ transforms the images from domain $A$ to domain $B$. We use $\widetilde{P}$ to denote the generative distribution, *i.e.*, $G_{AB}(X) \sim \widetilde{P}$ for $X \sim P$. Similarly, $\widetilde{Q}$ denotes the generative distribution of generator $G_{BA}$. The minimax game of CycleGAN is given by

$$\min_{G_{AB},G_{BA}} \max_{D_{AB},D_{BA}} \mathcal{L}_{\text{GAN}}^{AB}(G_{AB}, D_{AB}) + \mathcal{L}_{\text{GAN}}^{BA}(G_{BA}, D_{BA}) + \mathcal{L}_{\text{cyc}}(G_{AB}, G_{BA}) \qquad (3)$$

where GAN losses $\mathcal{L}_{\text{GAN}}^{AB}, \mathcal{L}_{\text{GAN}}^{BA}$ minimize the divergence between the generative distributions and the targets, and the cycle consistency loss $\mathcal{L}_{\text{cyc}}$ is used to regularize the generators (Zhu et al., 2017). Concretely, the GAN loss $\mathcal{L}_{\text{GAN}}^{AB}$ is defined as follows,

$$\mathcal{L}_{\text{GAN}}^{AB}(G_{AB}, D_{AB}) := \mathbb{E}_{x \sim Q}[\log D_{AB}(x)] + \mathbb{E}_{x \sim \widetilde{P}}[\log(1 - D_{AB}(x))]$$

Now we tilt the GAN losses in CycleGAN objective to guide the generators to perform controlled transfer. Specially, we replace $\mathcal{L}_{\text{GAN}}^{AB}$ in Eq. (3) with the orthogonal GAN loss $\mathcal{L}_{\text{OGAN}}^{AB}$ when updating the generator $G_{AB}$:

$$\mathcal{L}_{\text{OGAN}}^{AB}(G_{AB}, D_{AB}) := \mathbb{E}_{x \sim \widetilde{P}}[\log(1 - \phi(D_{AB}(x), r(x)))]$$

where $\phi(D_{AB}(x), r(x)) = \frac{D_{AB}(x)r(x)}{(1-D_{AB}(x))+D_{AB}(x)r(x)}$ and $r(x) = \frac{w_x \setminus w_1(x)_0}{w_x \setminus w_1(x)_1}$. Consider an extreme case where we allow the model to change all latents, *i.e.*, $z_1(x) = x$. As a result, $\mathcal{L}_{\text{OGAN}}$ degenerates to the original $\mathcal{L}_{\text{GAN}}$ since $w_x \setminus w_1 \equiv \frac{1}{2}$ and $\phi(D_{AB}(x), r(x)) \equiv D_{AB}(x)$. The other orthogonal GAN loss $\mathcal{L}_{\text{OGAN}}^{BA}$ can be similarly derived. For a given generator $G_{AB}$, the discriminator's optimum is achieved at $D_{AB}^*(x) = Q(x)/(\widetilde{P}(x) + Q(x))$. Assuming the discriminator is always trained to be optimal, the generator $G_{AB}$ can be viewed as optimizing the following virtual objective: $\mathcal{L}_{\text{OGAN}}^{AB}(G_{AB}) := \mathcal{L}_{\text{OGAN}}^{AB}(G_{AB}, D_{AB}^*) = \mathbb{E}_{x \sim \widetilde{P}} \log \frac{\widetilde{P}(x)}{\widetilde{P}(x)+Q(x)r(x)}$. The optimal generator in the new objective satisfies the following property.

**Proposition 2.** *The global minimum of $\mathcal{L}_{\text{OGAN}}^{AB}(G_{AB})$ is achieved if and only if $\widetilde{P}_{Z_1,Z_2}(z_1, z_2) = Q_{Z_1}(z_1)P_{Z_2}(z_2)$.*

We defer the proof to Appendix B.4. The proposition states that the new objective $\mathcal{L}_{\text{OGAN}}^{AB}$ converts the original global minimum $Q_{Z_1}Q_{Z_2}$ in $\mathcal{L}_{\text{GAN}}^{AB}$ to the desired one $Q_{Z_1}P_{Z_2}$. The symmetric statement holds for $\mathcal{L}_{\text{OGAN}}^{BA}(G_{BA})$.

## 4.1 EXPERIMENTS

### 4.1.1 SETUP

**Datasets.** (a) *CMNIST*: We construct C(olors)MNIST dataset based on MNIST digits (LeCun & Cortes, 2005). CMNIST has two domains and 60000 images with size $(3, 28, 28)$. The majority of the images in each domain will have a digit color and a background color correspond to the domain: domain A ↔ "red digit/green background" and domain B ↔ "blue digit/brown background". We define two *bias degree*s $\lambda_d, \lambda_b$ to control the probability of the images having domain-associated colors, *e.g.*, for an image in domain A, with $\lambda_d$ probability, its digit is set to be red; otherwise, its digit color is randomly red or blue. The background colors are determined similarly with parameter $\lambda_b$. In our experiments,

Table 1: CMNIST

|  | $Z_1$ acc. | $Z_2$ acc. |
|---|---|---|
| Vanilla | 90.2 | 14.3 |
| $w_x \smallsetminus w_1$ (MLDG) | 94.9 | 91.1 |
| $w_x \smallsetminus w_1$ (TRM) | 89.2 | 96.2 |
| $w_x \smallsetminus w_1$ (Fish) | 93.9 | 98.0 |
| IS (Oracle) | 90.0 | 100 |
| $w_x \smallsetminus w_1$ (Oracle) | 94.9 | 100 |

Table 2: CelebA-GH

|  | $Z_1$ acc. | $Z_2$ acc. | FID ↓ |
|---|---|---|---|
| Vanilla | 88.4 | 16.8 | **38.5** |
| $w_x \smallsetminus w_1$ (MLDG) | 90.9 | 53.0 | 46.2 |
| $w_x \smallsetminus w_1$ (TRM) | 92.2 | 56.5 | 39.8 |
| $w_x \smallsetminus w_1$ (Fish) | 88.8 | 42.9 | 43.4 |
| IS (Oracle) | 89.1 | 40.6 | 44.3 |
| $w_x \smallsetminus w_1$ (Oracle) | **93.7** | **57.4** | 39.7 |

we set $\lambda_d = 0.9$ and $\lambda_b = 0.8$. Our goal is to transfer the digit color ($Z_1$) while leaving the background color ($Z_2$) invariant. (b) *CelebA-GH*: We construct the CelebA-G(ender)H(air) dataset based on the gender and hair color attributes in CelebA (Liu et al., 2015). CelebA-GH consists of 110919 images resized to $(3, 128, 128)$. In CelebA-GH, domain A is non-blond-haired males, and the domain B is blond-haired females. Our goal is to transfer the facial characteristic of gender ($Z_1$) while keeping the hair color ($Z_2$) intact.

**Models.** We compare variants of orthogonal classifiers to the vanilla CycleGAN (**Vanilla**) and the importance sampling objective (**IS**). We consider two ways of obtaining the $w_1$ classifier: (a) *Oracle:* The principal classifier $w_1$ is trained on an oracle dataset where $Z_1$ is the only discriminative direction, *e.g.*, setting bias degrees $\lambda_d = 0.9$ and $\lambda_b = 0$ in CMNIST such that only digit colors vary across domains. (b) *Domain generalization:* Domain generalization algorithms aim to learn the prediction mechanism that is invariant across environments and thus generalizable to unseen environments (Blanchard et al., 2011; Arjovsky et al., 2019). The variability of environments is considered as nuisance variation. We construct environments such that only $Y|Z_1$ is invariant. We defer details of the environments to Appendix E.1. We use three domain generalization algorithms, Fish (Shi et al., 2021), TRM (Xu & Jaakkola, 2021) and MLDG (Li et al., 2018), to obtain $w_1$. We indicate different approaches by the parentheses after $w_x \smallsetminus w_1$ and IS, *e.g.*, $w_x \smallsetminus w_1$ (Oracle) is the orthogonal classifier with $w_1$ trained on Oracle datasets.

**Metric.** We use three metrics for quantitative evaluation: 1) $Z_1$ accuracy: the success rate of transferring an image's latent $z_1$ from domain A to domain B; 2) $Z_2$ accuracy: percentage of transferred images whose latents $z_2$ are unchanged; 3) FID scores: a standard metric of image quality (Heusel et al., 2017). We only report FID scores on the CelebA-GH dataset since it is not common to compute FID on MNIST images. $Z_1, Z_2$ accuracies are measured by two oracle classifiers that output an image's latents $z_1$ and $z_2$ (Appendix E.1.3).

For more details of datasets, models and training procedure, please refer to Appendix E.

### 4.1.2 RESULTS

**Comparison to IS.** In section 3.2, we demonstrate that IS suffers from high variance when the divergences of the marginal and the label-conditioned latent distributions are large. We provide further empirical evidence that our classifier orthogonalization method is more robust than IS. Fig. 2(a) shows that the test loss of IS increases dramatically with the divergences. Fig. 2(b) displays that IS's test loss grows rapidly when enlarging learning rate, which corroborates the observation that gradient variances are more detrimental to the model's generalization with larger learning rates (Wang et al., 2013). In contrast, the test loss of $w_x \smallsetminus w_1$ remains stable with varying divergences and learning rates. In Fig. 2(c), we visualize the histograms of predicted probabilities of $w_x \smallsetminus w_1$ (light color) and IS (dark color). We observe that the predicted probabilities of $w_x \smallsetminus w_1$ better concentrates around the ground-truth probabilities (red lines).

**Main result.** As shown in Table 1 and 2, adding an orthogonal classifier to the vanilla CycleGAN significantly improves its $z_2$ accuracy (from 14 to 90+ in CMNIST, from 16 to 40+ in CelebA-GH) while incurring a slight loss of the image quality. We observe that the oracle version of orthogonal classifier ($w_x \smallsetminus w_1$ (Oracle)) achieves best $z_2$ accuracy on both datasets. In addition, class orthogonalization is compatible with domain generalization algorithms when the prediction mechanism $Z_1|Y$ is invariant across collected environments.

In Fig. 3, we visualize the transferred samples from domain A. We observe that vanilla CycleGAN models change the background colors/hair colors along with digit colors/facial characteristics in CMNIST and CelebA. In contrast, orthogonal classifiers better preserve the orthogonal aspects $Z_2$ in the input images. We also provide visualizations for domain B in Appendix C.

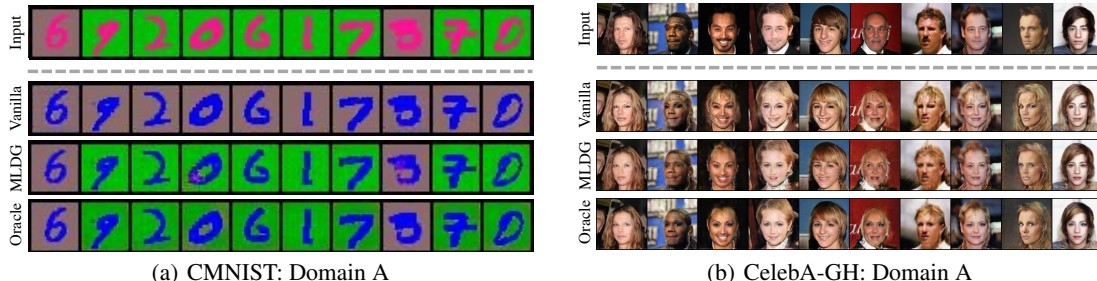

(a) CMNIST: Domain A       (b) CelebA-GH: Domain A

Figure 3: Style transfer samples on the domain A of CMNIST and CelebA-GH. We visualize the inputs (top row) and the corresponding transferred samples of different methods.

## 5   ORTHOGONAL CLASSIFIER FOR DOMAIN ADAPTATION

In unsupervised domain adaptation, we have two domains, source and target with data distribution $p_s$, $p_t$. Each sample comes with three variables, data $X$, task label $Y$ and domain label $U$. We assume $U$ is uniformly distributed in $\{0 \text{ (source)}, 1 \text{ (target)}\}$. The task label is only accessible in the source domain. Consider learning a model $h = g \circ f$ that is the composite of the encoder $f : \mathcal{X} \to \mathcal{Z}$ and the predictor $g : \mathcal{Z} \to \mathcal{Y}$. A common practice of domain adaptation is to match the two domain's feature distributions $p_s(f(X)), p_t(f(X))$ via domain adversarial training (Ganin & Lempitsky, 2015; Long et al., 2018; Shu et al., 2018). The encoder $f$ is optimized to extract useful feature for the predictor while simultaneously bridging the domain gap via the following domain adversarial objective:

$$\min_f \max_D \mathcal{L}(f, D) := \mathbb{E}_{x \sim p_s}[\log D(f(x))] + \mathbb{E}_{x \sim p_t}[\log(1 - D(f(x)))] \tag{4}$$

where discriminator $D : \mathcal{Z} \to [0, 1]$ distinguishes features from two domains. The equilibrium of the objective is achieved when the encoder perfectly aligns the two domain, i.e., $p_s(f(x)) = p_t(f(x))$. We highlight that such equilibrium is not desirable when the target domain has shifted label distribution (Azizzadenesheli et al., 2019; des Combes et al., 2020). Instead, when the label shift appears, it is more preferred to align the conditioned feature distribution, i.e., $p_s(f(x)|y) = p_t(f(x)|y), \forall y \in \mathcal{Y}$.

Now we show that our classifier orthogonalization technique can be applied to the discriminator for making it "orthogonal to" the task label. Specifically, consider the original discriminator as full classifier $w_x$, i.e., for a given $f$, $w_x(x)_0 = \arg\max_D \mathcal{L}(f, D) = \frac{p_s(f(x))}{p_s(f(x)) + p_t(f(x))}$. We then construct the principle classifier $w_1$ that discriminates the domain purely via the task label $Y$, i.e., $w_1(x)_0 = \Pr(U = 0 | Y = y(x))$. Note that here we assume the task label $Y$ can be determined by the data $X$, i.e., $Y = y(X)$. We focus on the case that $Y$ is discrete so $w_1$ could be directly computed via frequency count. We propose to train the encoder $f$ with the orthogonalized discriminator $w_x \setminus w_1$,

$$\min{}_f \mathcal{L}(f, (w_x \setminus w_1)(\cdot)_0) = \mathbb{E}_{x \sim p_s}[\log(w_x \setminus w_1)(x)_0] + \mathbb{E}_{x \sim p_t}[\log(1 - (w_x \setminus w_1)(x)_0)] \tag{5}$$

**Proposition 3.** *Suppose there exists random variable $Z_2 = z_2(X)$ orthogonal to $y(X) = Y$ w.r.t $p(f(X), U)$. Then $f$ achieves global optimum if and only if it aligns all label-conditioned feature distributions,* i.e., $p_s(f(x)|y) = p_t(f(x)|y), \forall y \in \mathcal{Y}$.

Note that in practice, we have no access to the target domain label prior $p_t(Y)$ and the label $y(x)$ for target domain data $x \sim p_t$. Thus we use the pseudo label $\hat{y}$ as a surrogate to construct the principle classifier, where $\hat{y}(x) = \arg\max_y h(x)_y$ is generated by our model $h$.

### 5.1   EXPERIMENTS

**Models.** We take a well-known domain adaptation algorithm **VADA** (Shu et al., 2018) as our baseline. VADA is based on domain adversarial training and combined with virtual adversarial training and entropy regularization. We show that utilizing orthogonal classifier can improve its robustness to label shift. We compare it with two improvements: (1) **VADA+$w_x \setminus w_1$** which is our method that plugs in the orthogonal classifier as a discriminator in VADA; (2) **VADA+IW**: We tailor the SOTA method for label shifted domain adaptation, importance-weighted domain adaptation (des Combes et al., 2020), and apply it to VADA. We also incorporate two recent domain adaptation algorithms for images— **CyCADA** (Hoffman et al., 2018) and **ADDA** (Tzeng et al., 2017)—into comparisons.

**Setup.** We focus on visual domain adaptation and directly borrow the datasets, architectures and domain setups from Shu et al. (2018). To add label shift between domains, we control the label distribution in the two domains. For source domain, we sub-sample $70\%$ data points from the first half of the classes and $30\%$ from the second half. We reverse the sub-sampling ratios on the target domain. The label distribution remains the same across the target domain train and test set. Please see Appendix E.2 for more details.

**Results.** Table 3 reports the test accuracy on seven domain adaptation tasks. We observe that VADA+$w_x \setminus w_1$ improves over VADA across all tasks and outperforms VADA+IW on five out of seven tasks. We find VADA+IW performs worse than VADA in two tasks, MNIST→SVHN and MNIST→MNIST-M. Our hypothesis is that the domain gap is large between these datasets, hindering the estimation of importance weight. Hence, VADA-IW is unable to adapt the label shift appropriately. In addition, the results show that VADA+$w_x \setminus w_1$ outperforms ADDA on six out of seven tasks.

Table 3: Test accuracy on visual domain adaptation benchmarks

| Source
Target | MNIST
MNIST-M | SVHN
MNIST | MNIST
SVHN | DIGITS
SVHN | SIGNS
GTSRB | CIFAR
STL | STL
CIFAR |
|---|---|---|---|---|---|---|---|
| Source-Only | 51.8 | 75.7 | 34.5 | 85.0 | 74.7 | 68.7 | 47.7 |
| ADDA | **89.7** | 78.2 | 38.4 | 86.0 | 90.6 | 66.8 | 50.4 |
| CyCADA | - | 82.8 | 39.6 | - | - | - | - |
| VADA | 77.8 | 79.0 | 35.7 | 90.3 | 93.6 | 72.4 | 53.1 |
| VADA + IW | 71.2↓ | 87.1↑ | 34.5 ↓ | **90.7** ↑ | **95.4**↑ | 74.0↑ | 53.8↑ |
| VADA + $w_x \setminus w_1$ | **79.1**↑ | **88.0**↑ | **40.5**↑ | 90.6↑ | 95.2↑ | **74.5** ↑ | **54.1**↑ |

## 6 ORTHOGONAL CLASSIFIER FOR FAIRNESS

We are given a dataset $\mathcal{D} = \{(x_t, y_t, u_t)\}_{t=1}^n$ that is sampled iid from the distribution $p_{XYU}$, which contains the observations $x \in \mathcal{X}$, the sensitive attributes $u \in \mathcal{U}$ and the labels $y \in \mathcal{Y}$. Our goal is to learn a classifier that is accurate w.r.t $y$ and fair w.r.t $u$. We frame the fairness problem as finding the orthogonal classifier of an "totally unfair" classifier $w_1$ that only uses the sensitive attributes $u$ for prediction. We can directly get the unfair classifier $w_1$ from the dataset statistics, *i.e.*, $w_1(x)_y = p(Y = y|U = u(x))$. Below we show that the orthogonal classifier of unfair classifier meets equalized odds, one metric for fairness evaluation.

**Proposition 4.** *If the orthogonal random variable of $U = u(X)$ w.r.t $p_{XY}$ exists, then the orthogonal classifier $w_x \setminus w_1$ satisfies the criterion of equalized odds.*

We emphasize that, unlike existing algorithms for learning fairness classifier, our method does not require additional training. We obtain a fair classifier via orthogonalizing an existing model $w_x$ which is simply the vanilla model trained by ERM on the dataset $\mathcal{D}$.

### 6.1 EXPERIMENTS

**Setup.** We experiment on the UCI Adult dataset, which has gender as the sensitive attribute and the UCI German credit dataset, which has age as the sensitive attribute (Zemel et al., 2013; Madras et al., 2018; Song et al., 2019). We compare the orthogonal classifier $w_x \setminus w_1$ to three baselines: **LAFTR** (Madras et al., 2018), which proposes adversarial objective functions that upper bounds the unfairness metrics; **L-MIFR** (Song et al., 2019), which uses mutual information objectives to control the expressiveness-fairness trade-off; **ReBias** (Bahng et al., 2020), which minimizes the Hilbert-Schmidt Independence Criterion between the model representations and biased representations; as well as the **Vanilla** ($w_x$) trained by ERM. We employ two fairness metrics – demographic parity distance ($\Delta_{\text{DP}}$) and equalized odds distance ($\Delta_{\text{EO}}$) – defined in Madras et al. (2018). We denote the penalty coefficient of the adversarial or de-biasing objective as $\gamma$, whose values govern a trade-off between prediction accuracy and fairness, in all baselines. We borrow experiment configurations, such as CNN architecture, from Song et al. (2019). Please refer to Appendix E.3 for more details.

**Results.** Tables 4, 5 report the test accuracy, $\Delta_{\text{DP}}$ and $\Delta_{\text{EO}}$ on Adults and German datasets. We observe that the orthogonal classifier decreases the unfairness degree of the Vanilla model and has competitive performances to existing baselines. Especially in the German dataset, compared to LAFTR, our method

has the same $\Delta_{\mathrm{EO}}$ but better $\Delta_{\mathrm{DP}}$ and test accuracy. It is surprising that our method outperforms LAFTR, even though it is not specially designed for fairness. Further, our method has benefit of being training-free, which allows it to be applied to improve any off-the-shelf classifier's fairness without additional training.

Table 4: Accuracy v.s. Fairness (Adults)

| | Acc.↑ | $\Delta_{\mathrm{DP}}$ ↓ | $\Delta_{\mathrm{EO}}$ ↓ |
|---|---|---|---|
| Vanilla | **84.5** | 0.19 | 0.20 |
| LAFTR ($\gamma = 0.1$) | 84.2 | 0.14 | 0.09 |
| LAFTR ($\gamma = 0.5$) | 84.0 | 0.12 | **0.07** |
| L-MIFR ($\gamma = 0.05$) | 81.6 | **0.04** | 0.15 |
| L-MIFR ($\gamma = 0.1$) | 82.0 | 0.06 | 0.16 |
| ReBias ($\gamma = 100$) | 84.3 | 0.15 | 0.11 |
| ReBias ($\gamma = 50$) | 84.4 | 0.17 | 0.16 |
| $w_x \smallsetminus w_1$ | 81.6 | 0.12 | 0.12 |

Table 5: Accuracy v.s. Fairness (German)

| | Acc.↑ | $\Delta_{\mathrm{DP}}$ ↓ | $\Delta_{\mathrm{EO}}$ ↓ |
|---|---|---|---|
| Vanilla | **76.0** | 0.19 | 0.33 |
| LAFTR ($\gamma = 0.1$) | 73.0 | 0.11 | **0.17** |
| LAFTR ($\gamma = 0.5$) | 72.7 | 0.11 | 0.19 |
| L-MIFR ($\gamma = 0.1$) | 75.8 | 0.10 | 0.21 |
| L-MIFR ($\gamma = 0.05$) | 75.6 | 0.08 | 0.18 |
| ReBias ($\gamma = 100$) | 73.0 | **0.07** | **0.17** |
| ReBias ($\gamma = 50$) | 75.0 | 0.10 | 0.20 |
| $w_x \smallsetminus w_1$ | 75.4 | 0.09 | 0.18 |

## 7 RELATED WORKS

**Disentangled representation learning** Similar to orthogonal random variables, disentangled representations are also based on specific notions of feature independence. For example, Higgins et al. (2018) defines disentangled representations via equivalence and independence of group transformations, Kim & Mnih (2018) relates disentanglement to distribution factorization, and Shu et al. (2020) characterizes disentanglement through generator consistency and a notion of restrictiveness. Our work differs in two key respects. First, most definitions of disentanglement rely primarily on the bijection between latents and inputs (Shu et al., 2020) absent labels. In contrast, our orthogonal features must be conditionally independent given labels. Further, in our work orthogonal features remain implicit and they are used discriminatively in predictors. Several approaches aim to learn disentangled representations in an unsupervised manner (Chen et al., 2016; Higgins et al., 2017; Chen et al., 2018). However, Locatello et al. (2019) argues that unsupervised disentangled representation learning is impossible without a proper inductive bias.

**Model de-biasing** A line of works focuses on preventing model replying on the dataset biases. Bahng et al. (2020) learns the de-biased representation by imposing HSIC penalty with biased representation, and Nam et al. (2020) trains an unbiased model by up-weighting the high loss samples in the biased model. Li et al. (2021) de-bias training data through data augmentation. However, these works lack theoretical definition for the de-biased model in general cases and often require explicit dataset biases.

**Density ratio estimation using a classifier** Using a binary classifier for estimating the density ratio (Sugiyama et al., 2012) enjoys widespread attention in machine learning. The density ratio estimated by classifier has been applied to Monte Carlo inference (Grover et al., 2019; Azadi et al., 2019), class imbalance (Byrd & Lipton, 2019) and domain adaptation (Azizzadenesheli et al., 2019).

**Learning a set of diverse classifiers** Another line of work related to ours is learning a collection of diverse classifiers through imposing penalties that relate to input gradients. Diversity here means that the classifiers rely on different sets of features. To encourage such diversity, Ross et al. (2018; 2020) propose a notion of local independence, which uses cosine similarity between input gradients of classifiers as the regularizer, while in Teney et al. (2021) the regularizer pertains to dot products. Ross et al. (2017) sequentially trains multiple classifiers to obtain qualitatively different decision boundaries. We defer more discussions of the advantages of classifier orthogonalization over existing methods to Appendix F.

## 8 CONCLUSION

We consider finding a discriminative direction that is orthogonal to a given principal classifier. The solution in the linear case is straightforward but does not generalize to the non-linear case. We define and investigate orthogonal random variables, and propose a simple but effective algorithm (classifier orthogonalization) to construct the orthogonal classifier with both theoretical and empirical support. Empirically, we demonstrate that the orthogonal classifier enables controlled style transfer, improves existing alignment methods for domain adaptation, and has a lower degree of unfairness.

ACKNOWLEDGEMENTS

The work was partially supported by an MIT-DSTA Singapore project and by an MIT-IBM Grand Challenge grant. YX was partially supported by the HDTV Grand Alliance Fellowship. We would like to thank the anonymous reviewers for their valuable feedback.

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

## A  MORE PROPERTIES OF ORTHOGONAL RANDOM VARIABLE

In this section we demonstrate the properties of orthogonal random variables. We also discuss the existence and uniqueness of orthogonal classifier when given the principal classifier. Below we list some useful properties of orthogonal random variables.

**Proposition 5.** *Let $Z_1$ and $Z_2$ be any mutually orthogonal w.r.t $P_{XY}$, then we have*

(i) **Invariance**: *For any diffeomorphism $g$, $g(Z_1)$ and $g(Z_2)$ are also mutually orthogonal.*

(ii) **Distribution-free**: *$Z_1, Z_2$ are mutually orthogonal w.r.t $P_{X'Y}$ if and only if there exists a diffeomorphism mapping between $X$ and $X'$.*

(iii) **Existence**: *Suppose the label-conditioned distributions $p_y \in \mathcal{C}^1$, $supp(p_y) = \mathcal{X}, \forall y \in \mathcal{Y}$. For $z_1 \in \mathcal{C}^1$ and $z_1(\mathcal{X})$ is diffeomorphism to Euclidean space, then $\forall i \in \mathcal{Y}$, there exists a diffeomorphism mapping $(z_1(X), z_{2,i}(X))$ such that $z_1(X) \perp\!\!\!\perp z_{2,i}(X)|Y = i$. Further if there exists diffeomorphism mappings between $z_{2,i}s$, then the orthogonal r.v. of $z_1(X)$ exists.*

In the following theorem, we justify the validity of the classifier orthogonalization when $z_1(x)$ satisfies some regularity conditions.

**Theorem 2.** *Suppose $w_1(x)_i = p(Y = i|z_1(x))$. If the label-condition distributions $p_y s$, and $z_1$ satisfy the conditions in Proposition 5.(iii), then $w_x \searrow w_1$ is the unique orthogonal classifier of $w_1$.*

Note that our construction of $w_x \searrow w_1$ does not require the exact expression of $z_1(X)$, its orthogonal r.v or data generation process. The theorem states that the orthogonal classifier constructed by classifier orthogonalization is the Bayesian classifier for any orthogonal random variables of $z_1(X)$.

## B  PROOFS

### B.1  PROOF FOR PROPOSITION 1

**Proposition 1.** *Suppose the orthogonal r.v. of $z_1(X)$ w.r.t $p_{XY}$ exists and is denoted as $z_2(X)$. Then $z(X) = z_2(X)$ is a maximizer of $I(z(X); Y|z_1(X))$ subject to $I(z(X); z_1(X)|Y) = 0$.*

*Proof.* For any function $z$, by chain rule we have $I(z_1(X), z_2(X); Y) = I(z_1(X); Y) + I(z_2(X); Y|z_1(X))$ and $I(z_1(X), z(X); Y) = I(z_1(X); Y) + I(z(X); Y|z_1(X))$. Further, the definition of the orthogonal r.v. ensures that there exists a differmorphism $f$ between $X$ and $z_1(X), z_2(X)$, which implies $I(z_1(X), z(f(z_1(X), z_2(X))); Y) = I(z_1(X), z(X); Y)$.

Hence by data processing inequality we have:

$$I(z_1(X), z_2(X); Y) \geq I(z_1(X), z(f(z_1(X), z_2(X))); Y) = I(z_1(X), z(X); Y)$$
$$\Leftrightarrow I(z_1(X); Y) + I(z_2(X); Y|z_1(X)) \geq I(z_1(X); Y) + I(z(X); Y|z_1(X))$$
$$\Leftrightarrow I(z_2(X); Y|z_1(X)) \geq I(z(X); Y|z_1(X))$$

The above inequality shows that $z = z_2$ maximize the mutual information $I(z(X); Y|z_1(X))$. Further, by definition of the orthogonal r.v., $z = z_2$ is independent of $z_1$ conditioned on $Y$, i.e. $z(X) \perp\!\!\!\perp z_1(X)|Y$ which is equivalent to $I(z(X); z_1(X)|Y) = 0$.  □

### B.2  PROOF FOR THEOREM 1

**Theorem 1.** *Assume $p_y$ is uniform distribution, $\forall w_x \in \mathcal{W}$ takes values in $(m, 1-m)^C$ with $m \in (0, \frac{1}{2})$, and $1/p_{X|Y}(x|y) \in (0, \gamma) \subset (0, +\infty)$ holds for $\forall x \in \mathcal{X}, y \in \mathcal{Y}$. Then for any $\delta \in (0, 1)$ with probability at least $1 - \delta$, we have:*

$$|R(\hat{w}_x \searrow w_1) - R(w_x^* \searrow w_1)| \leq (1 + \gamma) \left( 2\mathfrak{R}_{|\mathcal{D}|}(\mathcal{W}) + 2\log\frac{1}{m} \sqrt{\frac{2\log\frac{1}{\delta}}{|\mathcal{D}|}} \right)$$

*Proof.* Denote the empirical risk of $w$ as

$$\hat{R}(w) = \hat{R}(w) = -\frac{1}{|\mathcal{D}|} \sum_{(x_i, y_i) \in \mathcal{D}} \log w(x_i)_{y_i}$$

and the population risk as

$$R(w) = -\mathbb{E}_{p(x,y)}\left[\log w(x)_y\right]$$

*Step 1: bounding excess risk $|R(\hat{w}_x) - R(w_x^*)|$.*

By the classical result in theorem 8 in Bartlett & Mendelson (2001), we know that for any $w \in \mathcal{W}$, $\delta \in (0,1)$, since $-\log w(x)_y \in (0, \log\frac{1}{m})$, with probability at least $1 - \delta$,

$$|\hat{R}(w) - R(w)| = \Big|\frac{1}{n}\sum_{i=1}^{n}\log(w(x_i))_{y_i} - \mathbb{E}_p[\log(w(x)_y)]\Big| \le 2\mathfrak{R}_n(\mathcal{W}) + 2\log\frac{1}{m}\sqrt{\frac{2\log\frac{1}{\delta}}{n}}$$

Since $\hat{w}_x = \inf_{w_x \in \mathcal{W}}\hat{R}(w_x), w_x^* = \inf_{w_x \in \mathcal{W}}R(w_x)$, and $|\hat{R}(\hat{w}_x) - R(\hat{w}_x)| \le 2\mathfrak{R}_n(\mathcal{W}) + 2\log\frac{1}{m}\sqrt{\frac{2\log\frac{2}{\delta}}{n}}, |\hat{R}(w_x^*) - R(w_x^*)| \le 2\mathfrak{R}_n(\mathcal{W}) + 2\log\frac{1}{m}\sqrt{\frac{2\log\frac{1}{\delta}}{n}}$, we have

$$|R(\hat{w}_x) - R(w_x^*)| \le 2\mathfrak{R}_n(\mathcal{W}) + 2\log\frac{1}{m}\sqrt{\frac{2\log\frac{1}{\delta}}{n}} \tag{6}$$

*Step 2: bounding $|R(\hat{w}_x \smallsetminus w_1) - R(w_x^* \smallsetminus w_1)|$ by $|R(\hat{w}_x) - R(w_x^*)|$.*

$$R(w \smallsetminus w_1) = -\mathbb{E}_{p(x,y)}\left[\log\frac{\frac{w(x)_y}{w_1(x)_y}}{\sum_{y'}\frac{w(x)_{y'}}{w_1(x)_{y'}}}\right] = R(w) + \mathbb{E}_{p(x,y)}\left[\log w_1(x)_y\sum_{y'}\frac{w(x)_{y'}}{w_1(x)_{y'}}\right]$$

Then we have:

$$|R(\hat{w}_x \smallsetminus w_1) - R(w_x^* \smallsetminus w_1)|$$
$$\le |R(\hat{w}_x) - R(w_x^*)| + \left|\mathbb{E}_{p(x,y)}\left[\log w_1(x)_y\sum_{y'}\frac{\hat{w}_x(x)_{y'}}{w_1(x)_{y'}}\right] - \mathbb{E}_{p(x,y)}\left[\log w_1(x)_y\sum_{y'}\frac{w_x^*(x)_{y'}}{w_1(x)_{y'}}\right]\right|$$
$$= |R(\hat{w}_x) - R(w_x^*)| + \left|\mathbb{E}_{p(x,y)}\left[\log\frac{\sum_{y'}\frac{\hat{w}_x(x)_{y'}}{w_1(x)_{y'}}}{\sum_{y'}\frac{w_x^*(x)_{y'}}{w_1(x)_{y'}}}\right]\right|$$
$$\le |R(\hat{w}_x) - R(w_x^*)| + \max\left(\left|\mathbb{E}_{p(x,y)}\left[\log\max_{y'}\frac{\frac{\hat{w}_x(x)_{y'}}{w_1(x)_{y'}}}{\frac{w_x^*(x)_{y'}}{w_1(x)_{y'}}}\right]\right|, \left|\mathbb{E}_{p(x,y)}\left[\log\min_{y'}\frac{\frac{\hat{w}_x(x)_{y'}}{w_1(x)_{y'}}}{\frac{w_x^*(x)_{y'}}{w_1(x)_{y'}}}\right]\right|\right)$$

We define the ratio $r(x,y) = \frac{\hat{w}_x(x)_y}{w_x^*(x)_y}$. We define $\underline{y}(x) := \arg\min_y r(x,y), \bar{y}(x) := \arg\max_y r(x,y)$. We have $r(x,\underline{y}(x)) \le \frac{\sum_{y'}\hat{w}_x(x)_{y'}/w_1(x)_{y'}}{\sum_{y'}w_x^*(x)_{y'}/w_1(x)_{y'}} \le r(x,\bar{y}(x))$. Let assume $\frac{1}{p(y|x)} \le \gamma$ for all $x,y$. For any function $y' : \mathcal{X} \to \mathcal{Y}$, we have the following bound, where we have importance weight $\phi(x,y) := \frac{1}{p(y|x)}$ if $y = y'(x)$ otherwise 0,

$$|\mathbb{E}_x\log r(x,y'(x))| = |\mathbb{E}_{x,y}\phi(x,y)\log r(x,y)| \le \gamma|\mathbb{E}_{x,y}\log r(x,y)| = \gamma|R(\hat{w}_x) - R(w_x^*))|$$

As a result, we have

$$\left|\mathbb{E}\log\frac{\sum_{y'}\hat{w}_x(x)_{y'}/w_1(x)_{y'}}{\sum_{y'}w_x^*(x)_{y'}/w_1(x)_{y'}}\right| \le \max\left(|\mathbb{E}\log r(x,\underline{y}(x))|, |\mathbb{E}\log r(x,\bar{y}(x))|\right) \le \gamma|R(\hat{w}_x) - R(w_x^*)|$$

Thus

$$|R(\hat{w}_x \smallsetminus w_1) - R(w_x^* \smallsetminus w_1)| \le |R(\hat{w}_x) - R(w_x^*)|$$
$$+ \max\left(\left|\mathbb{E}_{p(x,y)}\left[\log\max_{y'}\frac{\frac{\hat{w}_x(x)_{y'}}{w_1(x)_{y'}}}{\frac{w_x^*(x)_{y'}}{w_1(x)_{y'}}}\right]\right|, \left|\mathbb{E}_{p(x,y)}\left[\log\min_{y'}\frac{\frac{\hat{w}_x(x)_{y'}}{w_1(x)_{y'}}}{\frac{w_x^*(x)_{y'}}{w_1(x)_{y'}}}\right]\right|\right)$$
$$\le (1 + \gamma)|R(\hat{w}_x) - R(w_x^*)| \tag{7}$$

Combining Eq. (6) and Eq. (7), we know that with probability $1 - \delta$, we have:

$$|R(\hat{w}_x \smallsetminus w_1) - R(w_x^* \smallsetminus w_1)| \le (1 + \gamma) \left( 2\mathfrak{R}_n(\mathcal{W}) + 2\log\frac{1}{m}\sqrt{\frac{2\log\frac{1}{\delta}}{n}} \right)$$

$\square$

### B.3 PROOF FOR PROPOSITION 5 AND THEOREM 2

**Proposition 5.** *Let $Z_1$ and $Z_2$ be any mutually orthogonal w.r.t $P_{XY}$, then we have*

(i) ***Invariance****: For any diffeomorphism $g$, $g(Z_1)$ and $g(Z_2)$ are also mutually orthogonal.*

(ii) ***Distribution-free****: $Z_1, Z_2$ are mutually orthogonal w.r.t $P_{X'Y}$ if and only if there exists a diffeomorphism mapping between $X$ and $X'$.*

(iii) ***Existence****: Suppose the label-conditioned distributions $p_y \in \mathcal{C}^1$, $supp(p_y) = \mathcal{X}, \forall y \in \mathcal{Y}$. For $z_1 \in \mathcal{C}^1$ and $z_1(\mathcal{X})$ is diffeomorphism to Euclidean space, then $\forall i \in \mathcal{Y}$, there exists a diffeomorphism mapping $(z_1(X), z_{2,i}(X))$ such that $z_1(X) \perp\!\!\!\perp z_{2,i}(X)|Y = i$. Further if there exists diffeomorphism mappings between $z_{2,i}s$, then the orthogonal r.v. of $z_1(X)$ exists.*

*Proof.* *(i)* Since $Z_1$ and $Z_2$ are independent, the independence also holds for $g(Z_1)$ and $g(Z_2)$. In addition, we construct the diffeomorphism between $(g(Z_1), g(Z_2))$ and $X$ as $\hat{f}(g(Z_1), g(Z_2)) = f(g^{-1}(g(Z_1)), g^{-1}(g(Z_2))) = X$. Apparently $f(g^{-1}(Z_1), g^{-1}(Z_2))$ is a diffeomorphism.

*(ii)* We denote the diffeomorphism between $X'$ and $X$ as $\hat{f}$. Then the diffeomorphism mapping between $(Z_1, Z_2)$ and $X$ is $\hat{f} \circ f$. Thus $Z_1, Z_2$ are mutually orthogonal w.r.t $X'$.

*(iii)* We will use a constructive proof.

*Step 1*: Denote $\dim(\mathcal{X}) = d, \dim(z_1(\mathcal{X})) = k$, then we can prove that the manifold $\left[z_1(\mathcal{X}), \mathbb{R}^{d-k}\right]$ is diffeomorphism of $\mathcal{X}$. We denote the diffeomorphism as $f$, and denote $f(x) = [z_1(x), t(x)]$ where $z_1 : \mathcal{X} \to z_1(\mathcal{X}), t : \mathcal{X} \to \mathbb{R}^{d-k}$.

*Step 2*: For $X|Y = y$, let $F_j(t(x)_j \mid t(x)_1, \cdots, t(x)_{j-1}, z_1(x), 1) : \mathbb{R} \to [0,1], j \in \{1, \ldots, d-k\}$ be the CDF corresponding to the distribution of $t(x)_j \mid t(X)_1 = t(x)_1, \cdots, t(X)_{j-1} = t(x)_{j-1}, z_1(X) = z_1(x), Y = y$.

$$h(t(x); z_1(x))_1 = F_1(t(x)_1 \mid z_1(x), y)$$
$$h(t(x); z_1(x))_i = F_i(t(x)_i \mid t(x)_1, \cdots, t(x)_{i-1}, z_1(x), 1), i \in \{2, \ldots, d\}$$

The conditions $p \in \mathcal{C}^1, p(x) > 0, \forall x \in \mathcal{X}$ ensure that the PDF of $f(X)$ is also in $\mathcal{C}^1$ and takes values in $(0, \infty)$. Thus the above CDFs are all diffeomorphism mapping. By the inverse CDF theorem, for any $z_1(x)$, $h(t(X); z_1(X) = z_1(x))$ is a uniform distribution in $(0,1)^{d-k}$. Hence the random variable defined by the mapping $h$, *i.e.*, $z_{2,y}(X) = h(t(X); z_1(X))$, is independent of $z_1(X)$ given $Y = 1$.

In addition, by the construction of $z_{2,y}(x)$ we know that there exists a diffeomorphism $\hat{f}$ between $(z_1(\mathcal{X}), t(\mathcal{X}))$ and $(z_1(\mathcal{X}), z_{2,y}(\mathcal{X}))$ such that $\hat{f}(z_1(X), t(X)) = (z_1(X), z_{2,y}(X))$. Then $f^{-1} \circ \hat{f}^{-1}(z_1(X), z_{2,y}(X)) = X$ is also a diffeomorphsim mapping.

*Step 3*: From the condition we know that for every $y \in \mathcal{Y}$, there exists a diffeomorphism function $m_y$ such that $m_y \circ z_{2,y}(x) = z_{2,1}(x)$. Apparently $\forall y, m_y \circ z_{2,y}(X) \perp\!\!\!\perp z_{2,1}(X)|Y = y$ by $z_{2,y}(X) \perp\!\!\!\perp z_1(X)|Y = y$. Further, $(z_1, z_{2,1})$ is a diffeomorphism. Hence $z_{2,1}(X)$ is the orthogonal r.v. of $z_1(X)$ w.r.t $P_{XY}$.

$\square$

**Theorem 2.** *Suppose $w_1(x)_i = p(Y = i|z_1(x))$. If the label-condition distributions $p_ys$, and $z_1$ satisfy the conditions in Proposition 5.(iii), then $w_x \smallsetminus w_1$ is the unique orthogonal classifier of $w_1$.*

*Proof.* By the existence property in Proposition 5, we know that for $X$, there exists a orthogonal r.v. of $z_1(X)$ and denote it as $z_2(X_1)$. By the proposition we know that there exists a common diffeomorphism $f$ mapping $(z_1(X|y), z_2(X|y))$ to $X|y, \forall y \in \mathcal{Y}$.

Further, by change of variables and the definition of orthogonal r.v., we have $\frac{p_{i,2}(z_2)}{p_{j,2}(z_2)} = \frac{p_{i,1}(z_1)p_{i,2}(z_2)\text{vol}J_f(z_1,z_2)}{p_{j,1}(z_1)p_{j,2}(z_2)\text{vol}J_f(z_1,z_2)} / \frac{p_{i,1}(z_1)}{p_{j,1}(z_1)} = \frac{p_i(x)}{p_j(x)} / \frac{p_{i,1}(z_1)}{p_{j,1}(z_1)} = \frac{w_x(x)_i}{w_x(x)_j} / \frac{w_1(x)_i}{w_1(x)_j}$. Thus via the bijection of classifier and density ratios we know that $w_x \smallsetminus w_1(x)_i = p(Y = i|z_2(x))$. □

## B.4 PROOF FOR PROPOSITION 2

**Proposition 2.** *The global minimum of $\mathcal{L}_{OGAN}^{AB}(G_{AB})$ is achieved if and only if $\widetilde{P}_{Z_1,Z_2}(z_1,z_2) = Q_{Z_1}(z_1)P_{Z_2}(z_2)$.*

*Proof.* By definition, $r(x) = \frac{w_2(x)}{1-w_2(x)} = \frac{P(z_2(x))}{Q(z_2(x))}$. Thus we can reformulate the criterion $\mathcal{L}_{OGAN}^{AB}$ as the following:

$$\mathcal{L}_{OGAN}^{AB}(G_{AB}) = \mathbb{E}_{x\sim\widetilde{P}}\log\frac{\widetilde{P}(x)}{\widetilde{P}(x) + Q(x)r(x)}$$

$$= \mathbb{E}_{z_1,z_2\sim\widetilde{P}}\log\frac{\widetilde{P}(z_1,z_2)}{\widetilde{P}(z_1,z_2) + Q(z_1)Q(z_2)\frac{P(z_2)}{Q(z_2)}}$$

$$\geq -\log\mathbb{E}_{z_1,z_2\sim\widetilde{P}}\Big[\frac{\widetilde{P}(z_1,z_2) + Q(z_1)P(z_2)}{\widetilde{P}(z_1,z_2)}\Big] = -\log 2$$

where we get the lower bound by Jensen's inequality. The equality holds when $\widetilde{P}(z_1,z_2) = Q(z_1)P(z_2)$. □

## B.5 PROOF FOR PROPOSITION 3

**Proposition 3.** *Suppose there exists random variable $Z_2 = z_2(X)$ orthogonal to $y(X) = Y$ w.r.t $p(f(X),U)$. Then $f$ achieves global optimum if and only if it aligns all label-conditioned feature distributions, i.e., $p_s(f(x)|y) = p_t(f(x)|y), \forall y \in \mathcal{Y}$.*

*Proof.* By definition, optimal classifier $w_x(x)_0 = \frac{p_s(f(x))}{p_s(f(x))+p_t(f(x))}$, principle classifier $w_1(x)_0 = \frac{p_s(y(x))}{p_s(y(x))+p_t(y(x))}$. By the definition of orthogonal r.v., $(Y,U)$ and $f(X)$ have a bijection between them. It suggests, conditioned on $Y$, the supports of $f(X)|Y = y$ is non-overlapped for different $y$ in both domains $p_s$ and $p_t$. It means if $p_s(f(x)|y) > 0$ then $\forall y' \neq y, p_s(f(x)|y') = 0$.

Thus the orthogonal classifier $w_x \smallsetminus w_1$ satisfies:

$$(w_x \smallsetminus w_1)(x)_0 = \frac{p_s(f(x))}{p_s(y(x))} / \Big(\frac{p_s(f(x))}{p_s(y(x))} + \frac{p_t(f(x))}{p_t(y(x))}\Big)$$

$$= \frac{\sum_{y'} p_s(f(x)|y')p_s(y')}{p_s(y(x))} \Big/ \Big(\frac{\sum_{y'} p_s(f(x)|y')p_s(y')}{p_s(y(x))} + \frac{\sum_{y'} p_t(f(x)|y')p_t(y')}{p_t(y(x))}\Big)$$

$$= \frac{p_s(f(x)|y(x))p_s(y(x))}{p_s(y(x))} \Big/ \Big(\frac{p_s(f(x)|y(x))p_s(y(x))}{p_s(y(x))} + \frac{p_t(f(x)|y(x))p_t(y(x))}{p_t(y(x))}\Big)$$

$$= \frac{p_s(f(x)|y(x))}{p_s(f(x)|y(x)) + p_t(f(x)|y(x))}$$

Thus the objective in Eq. (5) can be reformulated as,

$$\mathcal{L}(f, (w_x \smallsetminus w_1)(\cdot)_0)$$

$$= \mathbb{E}_{x\sim p_s}\Big[\log\frac{p_s(f(x)|y(x))}{p_s(f(x)|y(x)) + p_t(f(x)|y(x))}\Big] + \mathbb{E}_{x\sim p_t}\Big[\log\frac{p_t(f(x)|y(x))}{p_s(f(x)|y(x)) + p_t(f(x)|y(x))}\Big]$$

$$= \mathbb{E}_{y\sim p_s}\mathbb{E}_{x\sim p_s(x|y)}\Big[-\log\frac{p_s(f(x)|y) + p_t(f(x)|y)}{p_s(f(x)|y)}\Big] + \mathbb{E}_{y\sim p_t}\mathbb{E}_{x\sim p_t(x|y)}\Big[-\log\frac{p_s(f(x)|y) + p_t(f(x)|y)}{p_t(f(x)|y)}\Big]$$

$$\geq -\mathbb{E}_{y\sim p_s}\Big[\log\mathbb{E}_{x\sim p_s(x|y)}\Big[\frac{p_s(f(x)|y) + p_t(f(x)|y)}{p_s(f(x)|y)}\Big]\Big] - \mathbb{E}_{y\sim p_t}\Big[\log\mathbb{E}_{x\sim p_t(x|y)}\Big[\frac{p_s(f(x)|y) + p_t(f(x)|y)}{p_t(f(x)|y)}\Big]\Big]$$

$$= -2\log 2.$$

The lower-bound holds due to Jensen's inequality. The equality is achieved if and only if $\frac{p_s(f(x)|y)}{p_t(f(x)|y)}$ is invariant to $x$, for all $y$, *i.e.*, $\forall y \in \mathcal{Y}, p_s(f(x)|y) = p_t(f(x)|y)$.

$\square$

### B.6 Proof for Proposition 4

**Proposition 4.** *If the orthogonal random variable of $U = u(X)$ w.r.t $p_{XY}$ exists, then the orthogonal classifier $w_x \smallsetminus w_1$ satisfies the criterion of equalized odds.*

*Proof.* We denote the orthogonal random variables as $z_2(X)$, and $w_x \smallsetminus w_1(x) = \Pr[y|z_2(x)]$. Since $z_2(X) \perp\!\!\!\perp U|Y$, we know that $w_x \smallsetminus w_1(X) \perp\!\!\!\perp U|Y$. Hence the prediction of the classifier $w_x \smallsetminus w_1$ is conditional independent of sensitive attribute $U$ on the ground-truth label $Y$, which meets the equalized odds metric. $\square$

## C    EXTRA SAMPLES

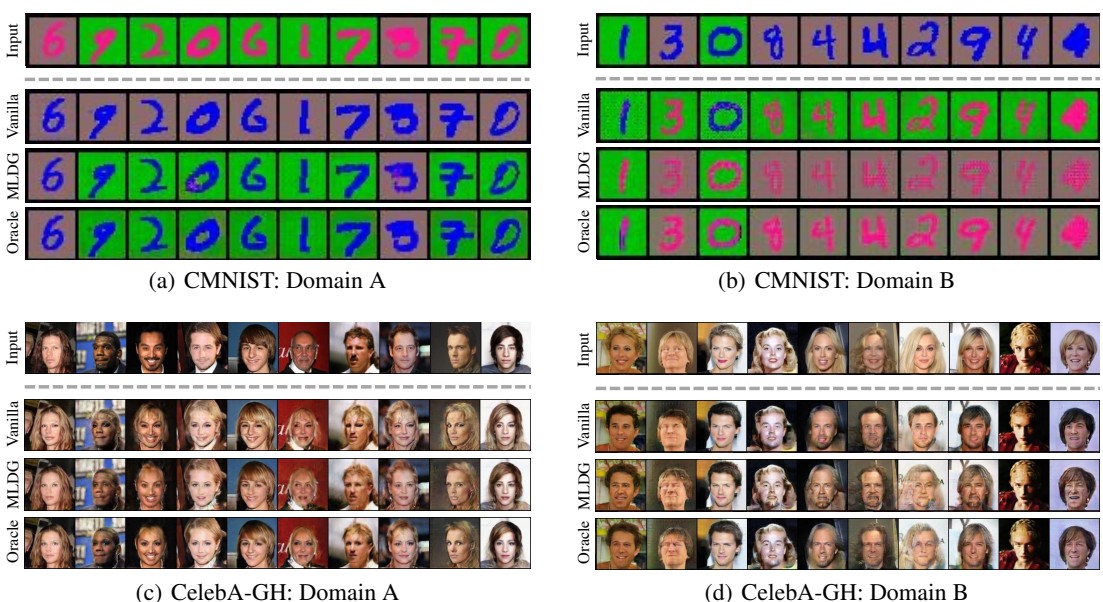

(a) CMNIST: Domain A

(b) CMNIST: Domain B

(c) CelebA-GH: Domain A

(d) CelebA-GH: Domain B

Figure 4: CMNIST and CelebA-GH

## D    EXTRA EXPERIMENTS

### D.1    STYLE TRANSFER

Table 6: CelebA

|  | $Z_1$ acc. | $Z_2$ acc. | FID $\downarrow$ |
|---|---|---|---|
| Vanilla | 75.3 | 87.0 | 36.2 |
| $w_x \smallsetminus w_1$-MLDG | 81.8 | 93.5 | 35.2 |
| $w_x \smallsetminus w_1$-TRM | 76.3 | 92.5 | 33.5 |
| $w_x \smallsetminus w_1$-Fish | 74.2 | 93.3 | 33.1 |
| IS-Oracle | 73.3 | 88.1 | 36.5 |
| $w_x \smallsetminus w_1$-Oracle | **84.0** | **95.2** | 38.5 |

We show transferred samples for both domains in Fig. 4 on CMNIST and CelebA-GH.

## D.2 STYLE TRANSFER ON CELEBA

Unlike the split in CelebA-GH dataset, domain A is males, and domain B is females. The two domains together form the full CelebA (Liu et al., 2015) with 202599 images. Note that there exists large imbalance of hair colors within the male group, with only around $2\%$ males having blond hair. Table 6 reports the $Z_1/Z_2$ accuracies and FID scores on the full CelebA datasets. We find that our method improves the style transfer on the original CelebA dataset, especially the $Z_2$ accuracy. It shows that our method can help style transfer in real-world datasets with multiple varying factors.

# E EXTRA EXPERIMENT DETAILS

## E.1 STYLE TRANSFER

### E.1.1 DATASET DETAILS

We randomly sampled two digits colors and two background colors for CMNIST. For CelebA-GH, we extract two subsets—male with non-blond hair and female with blond hair—by the attributes provided in CelebA dataset. For all datasets, we use 0.8/0.2 proportions to split the train/test set.

For CelebA dataset, following the implementation of Zhu et al. (2017) (`https://github.com/junyanz/CycleGAN`), we use 128-layer UNet as the generator and the discriminator in PatchGAN. For CMNIST dataset, we use a 6-layer CNN as the generator and a 5-layer CNN as the discriminator. We adopt Adam with learning rate 2e-4 as the optimizer and batch size 128/32 for CMNIST/CelebA.

### E.1.2 DETAILS OF MODELS

We craft datasets for training the oracle $w_1(x) = \Pr(Y|z_1(x))$. For CMNIST, oracle dataset has bias degrees 0.9/0. For CelebA-GH, oracle dataset has {male, female} as classes and the hair colors under each class are balanced.

For each dataset, we construct two environments to train the domain generalization models. For CMNIST, the two environments are datasets with bias degrees 0.9/0.4 and 0.9/0.8. For CelebA-GH, the two environments consist of different groups of equal size: one is {non-blond-haired males, blond-haired females} and the other is {non-blond-haired males, blond-haired females, non-blond-haired females, blond-haired males}. Note that the facial characteristic of gender ($Z_1$) is the invariant prediction mechanism across environments, instead of the hair color.

### E.1.3 EVALUATION FOR $Z_1/Z_2$ ACCURACY

To evaluate the accuracies on $Z_1$, $Z_2$ latents, we train two oracle classifiers $w_1^*(x) = \Pr(Y|z_1(x)), w_2^*(x) = \Pr(Y|z_2(x))$ on two designed datasets. Specifically, $w_1/w_2$ are trained on datasets $\mathcal{D}_1/\mathcal{D}_2$ with $Z_1/Z_2$ as the only discriminative direction. For CMNIST, $\mathcal{D}_1$ is the dataset with bias degrees 0.9/0 and $\mathcal{D}_2$ is the dataset with bias degrees 0/0.8. For CelebA-GH, $\mathcal{D}_1$ is the dataset with {male, female} as classes and the hair colors under each class are balanced. $\mathcal{D}_2$ is the dataset with {blond hair, non-blond hair} as classes and the males/females under each class are balanced.

The $Z_1$ accuracy on dataset $\mathcal{D}$ for transfer model $G$ is:

$$Z_1 \text{ accuracy} = \frac{1}{|\mathcal{D}|} \sum_{x \in \mathcal{D}} \mathbb{I}\left( \arg\max_y w_1^*(x) \neq \arg\max_y w_1^*(G(x)) \right)$$

And $Z_2$ accuracy is:

$$Z_1 \text{ accuracy} = \frac{1}{|\mathcal{D}|} \sum_{x \in \mathcal{D}} \mathbb{I}\left( \arg\max_y w_2^*(x) = \arg\max_y w_2^*(G(x)) \right)$$

where $\mathbb{I}$ is the indicator function. $Z_1$ accuracy measures the success rate of transferring the $Z_1$ and $Z_2$ accuracy measures the success rate of keeping $Z_2$ aspects unchanged.

## E.2 DOMAIN ADAPTATION

The full names for the seven datasets are MNIST, MNIST-M, SVHN, GTSRB, SYN-DIGITS (DIGITS), SYN-SIGNS (SIGNS), CIFAR-10 (CIFAR) and STL-10 (STL). We sub-sample each dataset by the designated imbalance ratio to construct the pairs.

We use the 18-layer neural network in Shu et al. (2018) with pre-process via instance-normalization. Following Shu et al. (2018), smaller CNN is used for digits (MNIST, MNIST-M, SVHN, DIGITS) and traffic signs (SIGNS, GTSRB) and larger CNN is used for CIFAR-10 and STL-10. We use the Adam optimizer and hyper-parameters in Appendix B in Shu et al. (2018) except for MNIST → MNIST-M, where we find that setting $\lambda_s = 1, \lambda_d = 1e-1$ largely improves the VADA performance over label shifts. Furthermore, we set $\lambda_d = 1e-2$, *i.e.*, the default value suggested by Shu et al. (2018), to enable adversarial feature alignment.

### E.3 FAIRNESS

The two datasets are: the UCI Adult dataset[2] which has gender as the sensitive attribute; the UCI German credit dataset[3] which has age as the sensitive attribute. We borrow the encoder, decoder and the final fc layer from Song et al. (2019). For fair comparison, we use the same encoder+fc layer as the function family $\mathcal{W}$ of our classifier since our method do not need decoder for reconstruction. We use Adam with learning rate 1e-3 as the optimizer, and a batch size of 64.

The original ReBias method trains a set of biased models to learn the de-biased representations (Bahng et al., 2020), such as designing CNNs with small receptive fields to capture the texture bias. However, in the fairness experiments, the bias is explicitly given, *i.e.*, the sensitive attribute. For fair comparison, we set the HSIC between the sensitive attribute and the representations as the de-bias regularizer.

## F LEARNING A DIVERSE SET OF CLASSIFIERS

In this section we discuss the difference and advantages of the classifier orthogonalization, compared to methods that learn a diverse set of classifiers. We also provide a counter-example to show that the orthogonal input gradient does not lead to a pair of orthogonal classifiers. We show that orthogonal classifiers might not in optimal solution set of minimizing input gradient penalty.

The goals of learning diverse classifiers include interpretability (Ross et al., 2017), overcoming simplicity bias (Teney et al., 2021), improving ensemble performance (Ross et al., 2020), and recovering confounding decision rules (Ross et al., 2018). There is a direct trade-off between diversity and accuracy and this is controlled by the regularization parameter. In contrast, in our work, given the principal classifier, our method directly constructs a classifier that uses only the orthogonal variables for prediction. There is no trade-off to set in this sense. We also focus on novel applications of controlling the orthogonal directions, such as style transfer, domain adaptation and fairness. Further, our notion of orthogonality is defined via latent orthogonal variables that relate to inputs via a diffeomorphism (Definition 1) as opposed to directly in the input space. Although learning diverse classifiers through input gradient penalties is efficient, it does not guarantee orthogonality in the sense we define it. We provide an illustrative counter-example in Appendix F, where the input space is not disentangled nor has orthogonal input gradients but corresponds to a pair of orthogonal classifiers in our sense. We also prove that, given the principal classifier, minimizing the loss by introducing an input gradient penalty would not necessarily lead to an orthogonal classifier.

We note that one clear limitation of our classifier orthogonalization procedure is that we need access to the full classifier. Learning this full classifier can be impacted by simplicity bias (Shah et al., 2020) that could prevent it from relying on all the relevant features. We can address this limitation by leveraging previous efforts (Ross et al., 2020; Teney et al., 2021) to mitigate simplicity bias when training the full classifier.

### F.1 RELATIONSHIP TO ORTHOGONAL INPUT GRADIENTS

The orthogonality constraints on input gradients demonstrate good performance in learning a set of diverse classifiers (Ross et al., 2017; 2018; 2020; Teney et al., 2021). They typically use the dot product of input gradients between pairs of classifiers as the regularizer. We highlight that when facing the latent orthogonal random variables, the orthogonal gradient constraints on input space no longer guarantees to learn the orthogonal classifier. We demonstrate it on a simple non-linear example below.

Consider a binary classification problem with the following data generating process. Label $Y$ is uniformly distributions in $\{-1, 1\}$. Conditioned on the label, we have two independent $k$-dimensional latents

---

[2]https://archive.ics.uci.edu/ml/datasets/adult
[3]https://archive.ics.uci.edu/ml/datasets/Statlog+%28German+Credit+Data%29

$Z_1, Z_2$ such that $Z_1|Y = y \sim \mathcal{N}(y\mu_1, I_k)$, $Z_2|Y = y \sim \mathcal{N}(y\mu_2, I_k)$. $\mu_1 \in \mathbb{R}^k, \mu_2 \in \mathbb{R}^m$ are the means of the latent variables, and $k > m$. The data $X$ is generated from latents via the following diffeomorphism $X = \begin{pmatrix} g(Z_1 - \begin{pmatrix} Z_2 \\ 0_{(k-m)} \end{pmatrix}) \\ Z_2 \end{pmatrix}$ where $g$ is an arbitrary non-linear diffeomorphism on $\mathbb{R}^k$.

We can see that $Z_1, Z_2$ are mutually orthogonal w.r.t the distribution $p_{XY}$. Now consider the Bayes optimal classifiers $w_1, w_2$ using variable $Z_1, Z_2$. We have $w_1(x) = p_{Y|Z_1}(1|z_1) = \sigma(2\mu_1^T z_1)$ and $w_2(x) = p_{Y|Z_2}(1|z_2) = \sigma(2\mu_2^T z_2)$ , where $\sigma$ is the sigmoid function. Apparently, they are a pair of orthogonal classifiers. Besides, we denote the classifier based on the first $m$ dimensions in $z_1$ for prediction as $w_3$, i.e., $w_3(x) = p_{Y|(Z_1)_{:m}}(1|(z_1)_{:m}) = \sigma(2(\mu_1)_{:m}^T (z_1)_{:m})$.

Given the principal classifier $w_1$, we consider using the loss function in Ross et al. (2018; 2020) to learn the orthogonal classifier of $w_1$, by adding a input gradient penalty term to the standard classification loss:

$$\mathcal{L}_{w_1}(w) = \mathcal{L}_c(w) + \lambda \mathbb{E}[\cos^2(\nabla_x w_1(x), \nabla_x w(x))]$$

$\mathcal{L}_c$ is the expected cross-entropy loss, cos is the cosine-similarity and $\lambda$ is the hyper-parameter for the gradient penalty term (Ross et al., 2018; 2020). Below we show that (1) the orthogonal classifier $w_2$ does not necessarily satisfy $\mathbb{E}[\cos^2(\nabla_x w_1(x), \nabla_x w_2(x))] = 0$. (2) the orthogonal classifier $w_2$ is not necessarily the minimizer of $\mathcal{L}_{w_1}(w)$.

**Proposition 6.** *For the above $(Z_1, Z_2)$, their corresponding Bayes classifiers $w_1, w_2$ satisfy $\mathbb{E}[\nabla_x w_1(x)^T \nabla_x w_2(x)] = 0$ if and only if $(\mu_1)_{:m}^T \mu_2 = 0$.*

*Proof.* Let $X_1 = g(Z_1 - \begin{pmatrix} Z_2 \\ 0_{(k-m)} \end{pmatrix})$, $X_2 = Z_2$ are the two components of $X$. Note that $z_1 = g^{-1}(x_1) + \begin{pmatrix} x_2 \\ 0_{(k-m)} \end{pmatrix}$. By chain rule, the input gradient of $w_1$ is

$$\nabla_x w_1(x) = \frac{dz_1}{dx} \nabla_{z_1} \Pr(Y = 1|Z_1 = z_1(x))$$

$$= \begin{pmatrix} \nabla_{x_1} g^{-1}(x_1) \\ I_{m \times k} \end{pmatrix}_{(k+m) \times k} \frac{e^{-2\mu_1^T z_1(x)}}{(1 + e^{-2\mu_1^T z_1(x)})^2} \cdot 2\mu_1$$

where $I_{m \times k}$ is the submatrix of $I_{k \times k}$. Similarly we get $\nabla_x w_2(x) = \begin{pmatrix} 0 \\ I_{m \times m} \end{pmatrix}_{(k+m) \times m} \frac{e^{-2\mu_2^T z_2(x)}}{(1 + e^{-2\mu_2^T z_2(x)})^2} \cdot 2\mu_2$.

Together, the dot product between input gradients is

$$\nabla_x w_1(x)^T \nabla_x w_2(x) = \frac{2e^{-2\mu_1^T z_1(x)}\mu_1^T}{(1 + e^{-2\mu_1^T z_1(x)})^2} \begin{pmatrix} \nabla_{x_1} g^{-1}(x_1) \\ I_{m \times k} \end{pmatrix}^T_{(k+m) \times k} \begin{pmatrix} 0 \\ I_{m \times m} \end{pmatrix}_{(k+m) \times m} \frac{2e^{-2\mu_2^T z_2(x)}\mu_2}{(1 + e^{-2\mu_2^T z_2(x)})^2}$$

$$= \frac{4e^{-2\mu_1^T z_1(x) - 2\mu_2^T z_2(x)}(\mu_1)_{:m}^T \mu_2}{(1 + e^{-2\mu_1^T z_1(x)})^2 (1 + e^{-2\mu_2^T z_2(x)})^2}$$

Since $\frac{4e^{-2\mu_1^T z_1(x) - 2\mu_2^T z_2(x)}}{(1 + e^{-2\mu_1^T z_1(x)})^2 (1 + e^{-2\mu_2^T z_2(x)})^2} > 0$, $\mathbb{E}[\nabla_x w_1(x)^T \nabla_x w_2(x)]$ if and only if $(\mu_1)_{:m}^T \mu_2 = 0$. $\qquad \square$

Proposition 6 first shows that the expected input gradient product of the two Bayes classifiers of underlying latents is non-zero, unless the linear decision boundaries on $z_1, z_2$ are orthogonal. Hence, given the principal classifier $w_1$, the expected dot product of input gradients between $w_1$ and its orthogonal classifier $w_2$ does not have to be zero.

**Proposition 7.** *The orthogonal classifer $w_2$ does not minimize $\mathcal{L}_{w_1}(w)$ if $(\mu_1)_{:m} = \mu_2$ and $\mu_1^T \nabla_{x_1} g^{-1}(x_1)^T \nabla_{x_1} g^{-1}(x_1)_{k \times m}(\mu_1)_{:m} < 0$. Particularly, we show that $\mathcal{L}_{w_1}(w_3) < \mathcal{L}_{w_1}(w_2)$ in this case.*

*Proof.* Let $w_3(x) = p_{Y|(Z_1)_{:m}}(1|(z_1)_{:m})$. The input gradient of $w_3(x)$ is

$$\nabla_x w_3(x) = (\nabla_x w_1(x))_{:m} = \frac{dz_1}{dx} \nabla_{(z_1)_{:m}} \Pr(Y = 1|(Z_1)_{:m} = (z_1(x))_{:m})$$

$$= \begin{pmatrix} (\nabla_{x_1} g^{-1}(x_1))_{k \times m} \\ I_{m \times m} \end{pmatrix}_{(k+m) \times m} \frac{e^{-2(\mu_1)_{:m}^T z_1(x)_{:m}}}{(1 + e^{-2(\mu_1)_{:m}^T z_1(x)_{:m}})^2} \cdot 2(\mu_1)_{:m}$$

When $(\mu_1)_{:m} = \mu_2$, the dot-product between $\nabla_x w_3(x), \nabla_x w_1(x)$ is

$$\nabla_x w_1(x)^T \nabla_x w_3(x) = \frac{2e^{-2\mu_1^T z_1(x)} \mu_1^T}{(1 + e^{-2\mu_1^T z_1(x)})^2} \begin{pmatrix} \nabla_{x_1} g^{-1}(x_1) \\ I_{m \times k} \end{pmatrix}^T_{(k+m) \times k}$$

$$\begin{pmatrix} (\nabla_{x_1} g^{-1}(x_1))_{k \times m} \\ I_{m \times m} \end{pmatrix}_{(k+m) \times m} \frac{e^{-2(\mu_1)_{:m}^T z_1(x)_{:m}}}{(1 + e^{-2(\mu_1)_{:m}^T z_1(x)_{:m}})^2} \cdot 2(\mu_1)_{:m}$$

$$= \frac{4e^{-2\mu_1^T z_1(x) - 2(\mu_1)_{:m}^T z_1(x)_{:m}}}{(1 + e^{-2\mu_1^T z_1(x)})^2 (1 + e^{-2(\mu_1)_{:m}^T z_1(x)_{:m}})^2} \left( \| \mu_2 \|_2^2 + \mu_1^T \nabla_{x_1} g^{-1}(x_1)^T \nabla_{x_1} g^{-1}(x_1)_{k \times m}(\mu_1)_{:m} \right)$$

When $(\mu_1)_{:m} = \mu_2$, we know that $z_1(x), z_2(x)$ are equally predictive of the label and thus $\mathcal{L}_c(w_3) = \mathcal{L}_c(w_2)$. Note that if $\mu_1^T \nabla_{x_1} g^{-1}(x_1)^T \nabla_{x_1} g^{-1}(x_1)_{k \times m}(\mu_1)_{:m} < 0$, we have $\mathbb{E}[\cos^2(\nabla_x w_1(x), \nabla_x w_2(x))] > \mathbb{E}[\cos^2(\nabla_x w_1(x), \nabla_x w_3(x))]$:

$$\mathbb{E}[\cos^2(\nabla_x w_1(x), \nabla_x w_2(x))]$$

$$= \mathbb{E}\left[ \left( \frac{\| \mu_2 \|_2^2}{\| \begin{pmatrix} \nabla_{x_1} g^{-1}(x_1) \\ I \end{pmatrix}_{(k+m) \times k} \mu_1 \|_2 \| \begin{pmatrix} 0 \\ I \end{pmatrix}_{(k+m) \times m} \mu_2 \|_2} \right)^2 \right]$$

$$> \mathbb{E}\left[ \left( \frac{\left( \| \mu_2 \|_2^2 + \mu_1^T \nabla_{x_1} g^{-1}(x_1)^T (\nabla_{x_1} g^{-1}(x_1))_{k \times m} \mu_2 \right)}{\| \begin{pmatrix} \nabla_{x_1} g^{-1}(x_1) \\ I \end{pmatrix}_{(k+m) \times k} \mu_1 \|_2 \| \begin{pmatrix} \nabla_{x_1} g^{-1}(x_1) \\ I_{m \times k} \end{pmatrix}_{(k+m) \times k} \mu_2 \|_2} \right)^2 \right]$$

$$= \mathbb{E}[\cos^2(\nabla_x w_1(x), \nabla_x w_3(x))]$$

The strict inequality holds by $\mu_1^T \nabla_{x_1} g^{-1}(x_1)^T \nabla_{x_1} g^{-1}(x_1)_{k \times m}(\mu_1)_{:m} < 0$

$$\| \begin{pmatrix} \nabla_{x_1} g^{-1}(x_1) \\ I_{m \times k} \end{pmatrix}_{(k+m) \times k} \mu_2 \|_2 = \| \begin{pmatrix} \nabla_{x_1} g^{-1}(x_1) \\ 0 \end{pmatrix}_{(k+m) \times k} \mu_2 + \begin{pmatrix} 0 \\ I_{m \times k} \end{pmatrix}_{(k+m) \times k} \mu_2 \|_2$$

$$= \left( \| \begin{pmatrix} \nabla_{x_1} g^{-1}(x_1) \\ 0 \end{pmatrix}_{(k+m) \times k} \mu_2 \|_2^2 + \| \begin{pmatrix} 0 \\ I_{m \times k} \end{pmatrix}_{(k+m) \times k} \mu_2 \|_2^2 \right)^{\frac{1}{2}}$$

$$\geq \| \begin{pmatrix} 0 \\ I_{m \times k} \end{pmatrix}_{(k+m) \times k} \mu_2 \|_2$$

To verify that $\mu_1^T \nabla_{x_1} g^{-1}(x_1)^T \nabla_{x_1} g^{-1}(x_1)_{k \times m}(\mu_1)_{:m} < 0$ does exist, pick $k = 3, m = 2$, $g^{-1}(x) = \begin{pmatrix} 2 & -3 & 3 \\ 1 & -1 & 5 \\ 1 & -1 & 1 \end{pmatrix} x$, and $\mu_1 = \begin{pmatrix} 1 \\ 1 \\ 1 \end{pmatrix}$. We have $\mu_1^T \nabla_{x_1} g^{-1}(x_1)^T \nabla_{x_1} g^{-1}(x_1)_{k \times m}(\mu_1)_{:m} = -2$ in this case.

Together, we have $\mathcal{L}_{w_1}(w_3) < \mathcal{L}_{w_1}(w_2)$.                    □

Proposition 7 shows that the orthogonal classifier $w_2$ is not the minimizer of the loss with input gradient penalty. The results above also hold for un-normalized version of input gradient penalty in Teney et al. (2021) by similar analysis.

