# OpenReview forum: "Controlling Directions Orthogonal to a Classifier"
_ICLR.cc/2022/Conference — ICLR 2022 Spotlight_

### Official Review · Reviewer_fQv4 · 2021-10-22

**Correctness:** 4
**Technical Novelty And Significance:** 3
**Empirical Novelty And Significance:** 3
**Recommendation:** 8
**Confidence:** 4

**Main Review:**

Strong (+) and weak (-) points
- (+) The authors propose and define a new concept: orthogonal classifiers.
- (+) The concept is general and justified/motivated by needs of seemingly disparate areas of research (style transfer, domain adaptation, fairness). This non-obvious connection between multiple areas is interesting in itself.
- (+) Demonstration of empirical benefits in multiple applications.
- (-) There are quite a few missing references to methods that also learn multiple complementary classifiers. So it's difficult to evaluate the novelty/advantage of the proposed method.
- (-) Imprecisions in the writing (see advice/comments below). I believe these could be fixed by the authors for the final version.

Detailed comments and advice for improvements (in the order of their occurrence in the paper):

- In the second paragraph of the introduction:
  - "joint distribution": maybe name it (P(X, Y))
  - the standard notation P(...) = (...) might be easier to read than (...) \sim (...)
  - "label-dependent" need a hyphen
  - "the orthogonal classifer (...) must satisfy": it sounds like an established concept, whereas this is something that you define for this paper. So, rather say: "we define the classifier w2 to be orthogonal if (...)"
  - Given a classifier w1, there are multiple orthogonal classifiers w2, but the current writing implies there is a single w2.

- The concepts of "principal classifier" and "principal variables" are used in the abstract and intro without being defined or given a reference about. Only Sec. 3 makes it clearer what they refer to. In the intro, maybe use a non-technical word like "a GIVEN classifier" ?
- Fig. 1: "principle", typo for principal ?
- Proposition 1: the first notation I(A,B) uses a comma, the second a semicolumn.
- Sec. 3: "Similar, ..." -> "similarLY"
- Sec. 3: you should define volJf right after its first use (before "Taken together"). The equation before "taken together" would also be easier to read with some spacing between the three terms.
- The equation after "taken together" is not numbered. Also add spacing between terms for readability.
- The following sentence has a grammatical issue. I don't understand its second part. What is the subject of "ensures" ?
"The validity of this procedure is subject to overlapping support of class-conditional distributions, ensures that wx;w1 remain non-zero."
- How can we make sure that this condition is satisfied ? Training a "full classifier" does seem trivial to me. Existing work has shown that neural networks can focus on only a few predictive features (see e.g. Pitfalls of Simplicity Bias in Neural Networks, Shah et al. 2020).
- "would be closeD" -> close
- Proposition 4: "exits" -> "exists" ?
- Missing references: Ross et al. and more recently Teney et al. also proposed methods to learn multiple classifiers that use the input data in different ways.
  - [Ensembles of Locally Independent Prediction Models](https://ojs.aaai.org//index.php/AAAI/article/view/6004)
  - [Learning Qualitatively Diverse and Interpretable Rules for Classification](https://arxiv.org/abs/1806.08716)
  - In this one, the authors train multiple classifiers sequentially to focus each on different input features.
[Right for the right reasons: Training differentiable models by constraining their explanations](https://arxiv.org/abs/1703.03717)
  - [Evading the Simplicity Bias](https://arxiv.org/abs/2105.05612)
  - The multiverse loss may also be related, although admittedly more distantly. It serves to increase the number of distinct discriminative directions of a learned representation by duplicating a cross-entropy loss over multiple linear classifiers with an orthogonality constraint on their weights.
  - [The multiverse loss for robust transfer learning](https://arxiv.org/abs/1511.09033)
  - [Maximal multiverse learning for promoting cross-task generalization of fine-tuned language models](https://aclanthology.org/2021.eacl-main.14/)
- The first key difference with disentanglement: "most disentanglement definitions do not take the diffeomorphism (...) into account". I'm not sure this is correct. At least colloquially, disentanglement is about "breaking down" observations into independent factors that drive the generative process, hence collectively give rise the observations. In other words, the space of the union of these disentangled factors do map, via a diffeomorphism, to the observation space. What do the authors think ? Is there really a "key difference" here ?
- "Several works aim to learn the disentangled representations": no need for "the"
- Missing word: "Density ratio estimation using A classifier"
- Missing word: "Using A binary classifier (...)"
- Conclusion: "the orthogonal classifier makes accurate predictions on the orthogonal aspects of the principal classifier." What does "aspect" mean ? I feel the sentence is not very useful as a conclusion because of using such a vague term.

**Summary Of The Paper:**

- The paper introduces the notion of "orthogonal classifiers": classifiers that rely on orthogonal variables. It starts with the simple linear case, and adds a definition that also applies to the non-linear case.
- The paper proposes two methods to identify a classifier orthogonal to a given one.
- It then describes 3 use cases: style transfer, domain adaptation, and fairness.


**Summary Of The Review:**

Preliminary recommendation: unsure, because of the missing references to existing methods that achieve something comparable.

Question to the authors:
- Can you comment on differences/advantages of their approach compared to existing ones ?
- Can you discuss the necessity/feasibility of learning a "full classifier" ?

**Edit after authors' response: I am satisfied with the precisions added by the authors to the manuscript and I strongly recommend it for acceptance.**

---

> ### Author Response · Authors · 2021-11-19
> **Thank you for your review and suggestions**
>
> Thank you for the detailed review and thoughtful feedback. Below we address specific questions.
>
> **Q: Can you comment on the differences/advantages of your approach; references to methods that also learn multiple complementary classifiers.**
>
> A: We have added the discussion in the related work section.
>
> The primary goal of our work is to construct a classifier that only uses the orthogonal variable for prediction, absent other accuracy and diversity trade-offs as in [1,2,4]. The concept of orthogonality is rigorously defined through label-conditioned independence and associated diffeomorphism between the input and latents. We also provide novel applications where controlling the orthogonal direction is particularly relevant, including style transfer, domain adaptation and fairness.
>
> Further, we define orthogonality relative to a pair of latent variables connected to observations via diffeomorphism (Definition 1) as opposed to operating in the input space. Although one can efficiently learn a set of diverse classifiers through orthogonal input gradients [1,2,3,4], it doesn’t guarantee an orthogonal classifier in our sense. For example, in Appendix F, we added a counter-example to demonstrate that a pair of orthogonal classifiers does not necessarily have zero expected cosine-similarity of input gradients. We also show that, given the principal classifier, minimizing the loss with input gradient penalty does not necessarily yield the orthogonal classifier. Our classifier orthogonalization procedure gives the orthogonal classifier in these cases.
>
> *[1] Ensembles of Locally Independent Prediction Models, Andrew Slavin Ross and Weiwei Pan and Leo Anthony Celi and Finale Doshi-Velez, 2020.*
>
> *[2] Learning Qualitatively Diverse and Interpretable Rules for Classification, Andrew Slavin Ross and Weiwei Pan and Finale Doshi-Velez, 2018.*
>
> *[3] Right for the Right Reasons: Training Differentiable Models by Constraining their Explanations,
> Andrew Slavin Ross and Michael C. Hughes and Finale Doshi-Velez, 2017.*
>
> *[4] Evading the Simplicity Bias: Training a Diverse Set of Models Discovers Solutions with Superior OOD Generalization, Damien Teney and Ehsan Abbasnejad and Simon Lucey and Anton van den Hengel, 2021.*
>
> **Q: Writing improvements.**
>
> A: Thank you for the corrections and comments. We have polished our writing according to your suggestions
>
> **Q: Necessity/feasibility of learning a "full classifier"? Training a "full classifier" does not seem trivial.  Existing work has shown that neural networks can focus on only a few predictive features (see e.g. Pitfalls of Simplicity Bias in Neural Networks, Shah et al. 2020).**
>
> A: Thank you for the great question. We agree that neural models run the risk of learning only partial relations, especially in the presence of simple cues (Shah et al. 2020). However, in the applications we consider, the full classifier is learned from relatively large numbers of examples (e.g., unlabeled source/target sets). Moreover, the two sets of variables (principle, orthogonal) in our settings do provide more information for the classifier thus likely incorporated.
>
> Let us further consider the necessity/feasibility of the full classifier under two different scenarios.
> - If the empirical “full classifier” achieves almost the optimal risk then the classifier, $\hat{w}_x = Pr(Y|g(X))$, where $g(X)$ are the associated learned features, is close to the ground truth full classifier $w_x = Pr(Y|X)$. The accuracy of features $g(X)$ is not specifically needed in our procedure, only the performance of $w_x$ as a classifier. Indeed, our PAC bound (Eq.(7), proof of Theorem 1, Appendix B.2) shows that the regret of the orthogonal classifier is bounded by the regret of the full classifier.
>
> - When the empirical “full classifier” fails and does not achieve close to optimal risk, the resulting bias is indeed harmful and hinders orthogonalization. But in this case, the bias stems from the NN architecture and the learning algorithm, influencing all other methods as well, ours included. E.g., learning orthogonal classifiers via the importance-sampling algorithm in section 3.2 would be similarly affected. One should try to mitigate the effect by improving the architecture, learning algorithm, or collecting more unbiased data.

---

> > ### Comment · Reviewer_fQv4 · 2021-11-19
> > **Almost satisfied with the additions to the paper**
> >
> > I appreciate the additional discussion in the related work. It introduced a number of typos however. See below.
> >
> > I also appreciate appendix F on the relation to input gradients.
> >
> > > We agree that neural models run the risk of learning only partial relations, especially in the presence of simple cues (Shah et al. 2020). However, in the applications we consider, the full classifier is learned from relatively large numbers of examples (e.g., unlabeled source/target sets).
> >
> > Are you suggesting that the large number of examples avoids learning "partial" classifiers ? Issues like the simplicity bias are unrelated to the amount of training data.
> >
> > > If the empirical “full classifier” achieves almost the optimal risk then the classifier (...)
> >
> > My concern with the simplicity bias and other effects can't be analyzed with the risk: when there are multiple predictive, redundant features, a trained neural network may only use some of them (still achieving minimum risk). I think this is what you acknowledge in your second bullet point. I think that the references in your comment ([4] and possibly [1]) do permit to overcome  this deficiency (with input gradients) whereas the method proposed here cannot. I think this is missing from the new discussion in the related work. They are different approaches with different definition of complementarity/orthogonality, and they all do allow different things.
> >
> > Let me state my understanding to make sure I am correct. My understanding is that the method does require a "full classifier" to be available, which uses both the principal and the orthogonal features. If the "simplicity bias" and other effects only allow to get a full classifier that uses some subset of all predictive features, then the proposed method will not be able to get a principal/orthogonal classifiers that use features among these subset. Is this correct ? If so, I think it would be useful to be upfront and state this requirement/limitation early in the paper (for the benefit of the reader). I'm leaning toward an recommendation for acceptance of the paper but I want to make it maximally-useful for the reader.
> >
> >
> > Typos:
> > > Although learning diverse classifier through input gradients penalty are efficient
> > classifierS
> > through AN input gradient penalty
> > *is* efficient
> >
> > > does not have orthogonal input gradient
> > *gradientS*

---

> > > ### Author Response · Authors · 2021-11-19
> > > **Thank you for the additional questions**
> > >
> > > Thank you for the additional questions. We will update our paper accordingly.
> > >
> > > Note that the purpose of our orthogonalization procedure is not to address simplicity bias. We use orthogonalization to explicitly control non-principal directions, e.g., what is aligned (domain adaptation, style transfer) or removing bias (fairness). Our orthogonalization procedure relies on the full classifier behaviorally, i.e., we use the predicted probability values, not needing access to how the predictions are realized internally.
> > >
> > > However, effectively, our full classifier should rely on both principal and orthogonal features. The reviewer is correct that our procedure can therefore be impacted by simplicity bias. As the reviewer points out, we can make use of techniques such as those in [1,4] to mitigate the impact of simplicity bias on the full classifier. [1,4] do not address our task but can lead to a better full classifier that our procedure takes as input.
> > >
> > > The full classifier may fail for many reasons, including simplicity bias. By referring to sample sizes, we meant to imply that a lack of data is not one of those reasons. In practice, the principal classifier is approximate, as is the full classifier, and the resulting orthogonal classifier will be approximate as well. That said, our experiments do show that the orthogonalization procedure works quite well.

---

> > > > ### Comment · Reviewer_fQv4 · 2021-11-19
> > > > **Satisfied with the precisions**
> > > >
> > > > OK Good to hear these precisions. I agree with all you say. My suggestion is to make it clear at the very beginning of the paper what the proposed algorithm takes in (a "full" classifier), what it gives out, and under what conditions (i.e. if the classifier is not "full" the resulting classifiers also won't cover all predictive features).
> > > >
> > > > Usually a "teaser" figure is a good place to convey this. But since there isn't any in this paper, I suppose it can be incorporated in the introduction. Personally, it took me too much reading (and the discussions above) to get this really clear. So I imagine that such an addition will be helpful to other readers.

---

> > > > > ### Author Response · Authors · 2021-11-22
> > > > > **Thank you for your suggestions**
> > > > >
> > > > > We have added more details about the algorithms in our introduction according to the reviewer's suggestions. We also included discussions of the simplicity bias in Section 3.1 and Section 7. The revisions are highlighted in red.

---

### Official Review · Reviewer_SmEk · 2021-11-01

**Correctness:** 4
**Technical Novelty And Significance:** 3
**Empirical Novelty And Significance:** 3
**Recommendation:** 8
**Confidence:** 3

**Main Review:**

Strengths:

1. The paper does an excellent job of defining orthogonal classifiers and provides a clean solution to learning these models.

2. The mapping of multiple different problems is extremely interesting and provides a strong example of how to apply the approach to different problem settings.

3. The proposed approach appears to yield improvements for both the controlled style transfer and domain adaptation problems.

Weaknesses:

1. The controlled style transfer problem seems somewhat contrived. Providing a reasonable example of where this would be applicable would significantly increase the value demonstrated in this section.

2. Given the controlled style transfer problem is novel, it's difficult to determine whether the empirical results are significant as the baseline compared against is designed for a non-controlled style transfer problem.

3. Comparisons to a wider range of domain adaptation approaches (e.g. Cycada, HIDC, etc.) would make the results more convincing.

4. Empirical comparisons to de-biasing techniques would significantly strengthen the experimental section.

5. The results for fairness are somewhat uninteresting as the proposed approach simply reduces to weighting of labels for discrete sensitive attributes.

**Summary Of The Paper:**

Given a defined notion of orthogonality for random variables, this paper proposes a straight-forward approach to constructing classifiers orthogonal to a given classifier. Examples of mapping problems to the orthogonal classifier setting are provided for domain adaptation and fairness as well as the newly proposed problem of controlled style transfer.

**Summary Of The Review:**

I am overall in favor of accepting this paper. The proposed formulation and solution are well presented and the application to different problems is interesting.

---

> ### Author Response · Authors · 2021-11-19
> **Thank you for your review and suggestions**
>
> Thank you for the detailed review and thoughtful feedback. Below we address specific questions.
>
> **Q: The controlled style transfer problem; applicability**
>
>
> A:  Thanks for the question. We indeed introduce a new style transfer problem where there are multiple confounding differences allowed between the domains. This is actually closer to any real-world scenario than previous style transfer formulations. For example, in speech transfer, style differences also typically involve different speakers (as confounding factors). In terms of images, available radiology images between normal and patient conditions typically also pertain to different individuals, thus confounded. To better study and highlight our challenge, we used controlled experiments on the MNIST-C and CelebA-GH datasets in the paper. Note that the primary focus of the paper is the concept of classifier orthogonalization, not style transfer specifically.
>
> Appendix D.2 also gives results on CelebA. Unlike in CelebA-GH, the two domains in CelebA are constructed solely by gender attributes. Domain A represents males while domain B represents females. The CelebA dataset has a large imbalance of hair colors among males with only around 2% of males having blond hair. The results show that our method improves the style transfer on the original CelebA dataset, especially the $Z_2$ accuracy.
>
> **Q: Novelty of the style transfer problem and the resulting lack of specific baselines.**
>
> A: It is true that we introduce a new task as well as a new solution to it based on classifier orthogonalization. Previous non-controlled style transfer methods are used as baselines to illustrate the need for advances. It is non-trivial to adapt previous non-controlled style transfer methods to handle controlled style transfer tasks. This speaks for the novelty of the task and the approach.
>
> **Q: Comparisons to more domain adaptation approaches (e.g. Cycada, HIDC, etc.) would make the results more convincing.**
>
> A: Thank you for your suggestions. While we agree that additional comparisons would help, we note that the focus of our paper is on classifier orthogonalization, not only domain adaptation. Our comparisons have shown that using the orthogonal classifier (VADA + $w_x\setminus w_1$) can help improve the vanilla approach (VADA). We have now included more domain adaptation approaches for vision tasks --- ADDA [1] and CyCADA [2] --- into Section 5. CyCADA requires training a CycleGAN between each pair of datasets, and empirically we find that it does not get high-quality transferred images. Given the time limitation, we only include these results for the MNIST$\leftrightarrow$SVHN pair as implemented in the available code (https://github.com/jhoffman/cycada_release). The results show that our method (VADA + $w_x\setminus w_1$) outperforms ADDA on six of seven tasks. Although CyCADA beats VADA on the MNIST$\leftrightarrow$SVHN pair, we find that VADA + $w_x\setminus w_1$ remains superior to CyCADA.
>
> *[1] Adversarial Discriminative Domain Adaptation, Eric Tzeng and Judy Hoffman and Kate Saenko and Trevor Darrell, 2017*
>
> *[2] CyCADA: Cycle-Consistent Adversarial Domain Adaptation, Judy Hoffman and Eric Tzeng and Taesung Park and Jun-Yan Zhu and Phillip Isola and Kate Saenko and Alexei A. Efros and Trevor Darrell, 2018*
>
> **Q: Empirical comparisons to de-biasing techniques would significantly strengthen the experimental section.**
>
> A: Thanks for the suggestions. We have included another de-biasing baseline, ReBias [3], in Sec 6. The results show that ReBias reduces the model's unfairness with a slight drop in accuracy. Our method has competitive $\Delta_{\textrm{EO}}$ and $\Delta_{\textrm{DP}}$, and a higher test accuracy than ReBias on the German dataset.
>
> *[3] Learning De-biased Representations with Biased Representations, Hyojin Bahng and Sanghyuk Chun and Sangdoo Yun and Jaegul Choo and Seong Joon Oh, 2020*
>
>
> **Q: The results for fairness, label weighting for discrete sensitive attributes.**
>
> A: Our goal was to demonstrate the concept and utility of orthogonalization on the fairness task. Despite the simplicity, our method can improve fairness of existing classifiers without resorting to additional training. We agree that in a more realistic scenario, the biased classifier $w_1$ will be more complicated for continuous sensitive attributes.

---

> > ### Comment · Reviewer_SmEk · 2021-11-30
> > **Satisfied with Response**
> >
> > Thank you to the authors for the detailed response, I'm satisfied with the response.

---

### Official Review · Reviewer_dNRJ · 2021-11-03

**Correctness:** 3
**Technical Novelty And Significance:** 3
**Empirical Novelty And Significance:** 3
**Recommendation:** 6
**Confidence:** 3

**Main Review:**

- Strength
  - The paper presents rather a simple, but principled and novel way of constructing orthogonal classifier for any non-linear classifiers.
  - The paper is written with a good flow.
  - The effectiveness of the proposed technique has been confirmed on various applications, including style transfer, domain adaptation, and fairness.

- Weakness
  - For domain adaptation experiments, it seems authors assume access to the marginal distribution of label variables for the test set. This seems additional information the proposed method is using on top of VADA. Would you please confirm? Also, what if these marginal distributions between domains are exactly the same? Are there going to be any improvement?
  - Is label distribution of the test set for domain adaptation experiments uniform or does it follow the distribution of the target domain used for training?
  - For domain adaptation experiments, one baseline could be post-processing the source-only model, similarly to the procedure for constructing fair classifier in Section 6.

**Summary Of The Paper:**

The paper presents classifier orthogonalization technique that works for non-linear classifiers. The idea is to find a method that orthogonalizes the full classifier w.r.t. the principal classifier so that the resulting classifier is statistically orthogonal to principal classifier. The algorithm turns out to be very simple, only requiring access to the full classifier $P(Y|x)$ and the principal classifier $P(Y|z(x))$, where $z$ is a control variable you want to orthogonalize against. The paper also discusses an alternative method of classifier orthogonalization using importance sampling.

The effectiveness of the proposed classifier orthogonalization technique has been demonstrated through 3 applications: controlled style transfer, domain adaptation, and classifier fairness. For controlled style transfer, authors modified the CycleGAN's generator update step by orthogonalizing discriminator w.r.t. the controlled style variable. For domain adaptation, they modified VADA, an adversarial domain adaptation approach, by orthogonalizing discriminator based on the label-based principal classifier, so that discriminator wouldn't discriminate the domain based on the frequency count of labels from each domain. For fairness, full classifier is orthogonalized by the sensitive attribute classifier.

**Summary Of The Review:**

Overall I find that the proposed method novel and principled. As paper has demonstrated the proposed method could have great impact multiple application domains. As mentioned in the review, some information could be verified through discussion, but I am leaning towards acceptance for my initial rating.

---

> ### Author Response · Authors · 2021-11-19
> **Thank you for your review and suggestions**
>
> Thank you for the detailed review and thoughtful feedback. Below we address specific questions.
>
> **Q: Regarding the marginal distribution of label variables.**
>
> A: Our method does not assume access to the marginal distribution of the task label variable. Instead, we build the principal classifier w1 using (estimated) marginal label distributions, utilizing the labeled source set and the unlabelled target set. Ideally, we would have access to the label marginal as it would allow us to directly build the principal classifier. However, in all of our experiments, we have no access to prior $p_t(Y)$ over the target domain nor labels themselves. To build the principal classifier, we use surrogate labels $\hat{y}$ obtained from a trained classifier $\hat{y}(x)=\arg\max_{y} h(x)_y$.
>
> When the marginal distributions of labels across the domains agree, the principal classifier would not be discriminative, i.e., it would not help distinguish the domains based on the task label. As a result, $w_1(x)_0=\frac{1}{2}$, and the full classifier remains unchanged after orthogonalization. Hence, in this scenario, VADA+$w_x\setminus w_1$ reduces to vanilla VADA. This formulation would not offer benefits absent label shift.
>
> **Q: Label distribution in training and test sets.**
>
> A: Thanks for the question. The test set label distribution follows the same distribution as the target domain. We have made clarifications in our revised version.
>
> **Q: For domain adaptation experiments, one baseline could be post-processing the source-only model, similarly to the procedure for a fair classifier in Section 6.**
>
> A: Thanks for your suggestion. Unfortunately, similar post-processing is not directly applicable in domain adaptation. In the context of fairness, classifier orthogonalization pertained to the label predictor so as to reduce bias. In contrast, in domain adaptation, the subject for classifier orthogonalization is the discriminator, not the label classifier itself. The discriminator is used to align features between the domains without altering label information.

---

> > ### Comment · Reviewer_dNRJ · 2021-11-30
> > **Thanks**
> >
> > Thank you for your response and clarification on experimental settings. I believe the paper is good for publication.

---

### Author Response · Authors · 2021-11-19
**A summary of updates**

We would like to thank all reviewers for high-quality reviews and constructive feedback. We have revised our draft according to these comments. Major revisions are highlighted in blue in the new version. Below we provide a brief summary of updates:

### 1. Improved discussion of related work & improved readability

We have polished the exposition in response to Reviewer dNRJ and Reviewer fQv4. In addition, we have added the missing references pointed out by Reviewer fQv4, and discussed the differences and advantages in comparison to our method (Section 7, Appendix F). Note that we had originally mistakenly included a latex package in the submission that accidentally resulted in larger than the standard font. This has also been fixed.

### 2. Additional experiments

As suggested by Reviewer SmEk, we have added baselines to the domain adaptation experiments (Section 5) as well as fairness (Section 6).

---

### Decision · Program_Chairs · 2022-01-20

**Decision:**

Accept (Spotlight)

**Comment:**

This paper introduces the concept of classifier orthogonalization. This is a generalization of orthogonality of linear classifiers (linear classifiers with orthogonal weights) to the non-linear setting. It introduces the notion of a full and principal classifier, where the full classifier is one that minimizes the empirical risk, and the principal classifier is one that uses only partial information. The orthogonalization procedure assumes that the input domain, X can be divided into two sets of latent random variables Z1 and Z2 via a bijective mapping. The random variables Z1 are the principal random variables, and Z2 contains all other information. Z1 and Z2 are assumed to be conditionally independent given the target label. The paper outlines two approaches to construct orthogonal classifiers that operate only on Z2. The approach is highlighted in three applications: controlled style transfer, domain adaptation, and fair classification.

The reviewers all found the proposed method to be principled and compelling. Beyond clarification questions and some discussion on related work, the reviewers raised a few issues that were subsequently addressed: 1) Additional baselines for domain adaptation and fairness. 2) Controlled style transfer being a new task with no established baselines, and 3) The feasibility of training a proper “full classifier” that minimizes the empirical risk, and its necessity in the approach. The authors addressed these concerns and updated the paper, to the satisfaction of the reviewers. All of them unanimously recommend acceptance.